# Look Back to Reason Forward: Revisitable Memory for Long-Context LLM Agents

**Yaorui Shi**[1†], **Yuxin Chen**[2†], **Siyuan Wang**[3], **Sihang Li**[1], **Hengxing Cai**[4],
**Qi Gu**[5], **Xiang Wang**[1‡], **An Zhang**[1‡]

[1] University of Science and Technology of China, [2] National University of Singapore
[3] Shanghai Jiao Tong University, [4] DP Technology, [5] Meituan
{yaoruishi, xiangwang1223}@gmail.com, an_zhang@ustc.edu.cn
[†] Equal Contribution. [‡] Corresponding author.

## Abstract

Large language models face challenges in long-context question answering, where key evidence of a query may be dispersed across millions of tokens. Existing works equip large language models with a memory buffer that is dynamically updated via a linear document scan, also known as the "memorize while reading" methods. While this approach scales efficiently, it suffers from pruning of latent evidence, information loss through overwriting, and sparse reinforcement learning signals. To tackle these challenges, we present ReMemR1, which integrates the mechanism of memory retrieval into the memory update process, enabling the agent to selectively callback historical memories for non-linear reasoning. To further strengthen training, we propose a multi-level reward design, which combines final-answer rewards with dense, step-level signals that guide effective memory use. Together, these contributions mitigate information degradation, improve supervision, and support complex multi-hop reasoning. Extensive experiments demonstrate that ReMemR1 significantly outperforms state-of-the-art baselines on long-context question answering while incurring negligible computational overhead, validating its ability to trade marginal cost for robust long-context reasoning. Our code is available at https://github.com/syr-cn/ReMemR1.

## 1 Introduction

Reasoning over vast, multi-document contexts remains a critical bottleneck for large language models (LLMs) (Hsieh et al., 2024; Team et al., 2024; Beltagy et al., 2020; Ding et al., 2023; Child et al., 2019). This capability is crucial for real-world applications, such as synthesizing legal precedents or reviewing scientific literature, where critical evidence for a single query can be scattered across millions of tokens. However, the quadratic complexity of attention mechanisms makes it difficult for LLMs to track long-range dependencies and faithfully synthesize disparate information into a coherent answer.

To mitigate this, two primary paradigms have emerged. The first is Full-Text Context Retrieval (Figure 1(a)), where a retriever fetches relevant chunks from a corpus to form a prompt (Jin et al., 2025; Song et al., 2025; Shi et al., 2025; Du et al., 2026; 2025a). While widely used, this approach presents the LLM with fragmented, partial information and suffers from a heavy storage burden for the vector index. Alternatively, recent research explores the "memorize while reading" paradigm (Yu et al., 2025a; Li et al., 2025; Wang et al., 2025b; Shi et al., 2026) to handle infinite contexts linearly. As shown in Figure 1(b), this framework employs a memory agent that digests documents sequentially. At each step, the agent consumes a document chunk $c_t$ together with its previous memory $m_t$ and compresses them into a new memory $m_{t+1}$. After a single linear pass through the entire document, the agent uses this final memory $m_T$ buffer to generate an answer for the given question. This reduces the complexity of long-context question answering to linear time.

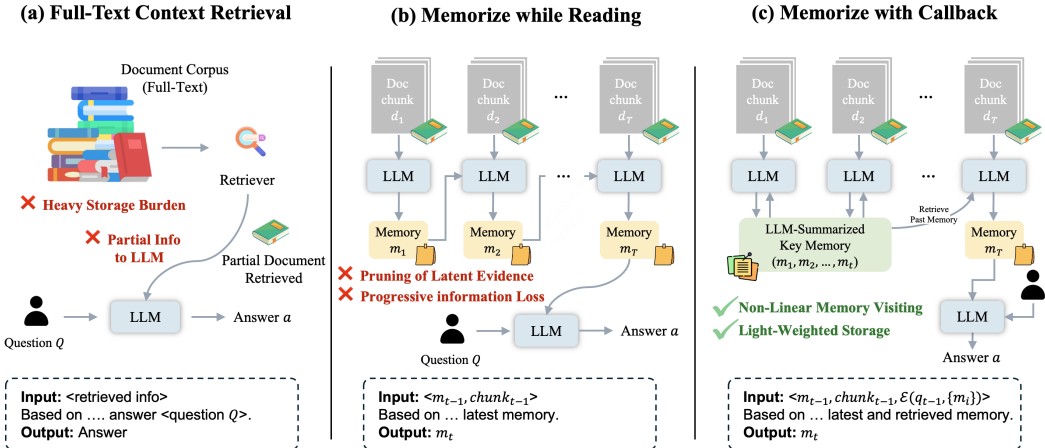

Figure 1: **Comparison of memory paradigms.** (a) Full-Text Retrieval separates retrieval from reasoning and incurs heavy storage burden. (b) "Memorize while Reading" paradigm suffers from progressive information loss and important information neglection due to linear memory overwriting. (c) This work introduces a callback mechanism, enabling non-linear memory visiting over past details and light-weighted storage.

Despite its efficiency, we identify the following intrinsic limitations in the existing "memorize while reading" paradigm:

- **Premature Pruning of Latent Evidence.** Standard memory agents evaluate the importance of the current document chunk $c_t$ based solely on the current memory state $m_t$. However, complex multi-hop reasoning could require integrating evidence found at different positions in a text. For instance, an agent might encounter a piece of evidence early on (Step $t$) whose significance only becomes apparent after reading a later section (Step $t + k$). Crucially, such limitation cannot be solved solely by improving the memory update policy, as the relevance of specific information may only become apparent with certain prior knowledge, which can be embedded in future context.

- **Progressive Information Loss in Memory Overwriting.** The paradigm's reliance on a fixed-length memory buffer necessitates constant information compression. This progressive degradation of memory makes it difficult to maintain the full context and impedes the ability to resolve complex queries that require synthesizing evidence spread across distant sections of the document.

- **Sparse and Delayed Supervision.** Training these agents using reinforcement learning typically relies on a single reward signal, such as the correctness of the final answer. This sparse reward at the end of the reasoning process offers limited guidance for the long sequence of intermediate memory updates, leading to inefficient optimization and suboptimal memory management strategies, particularly in complex tasks where producing correct final answers is especially challenging.

To address these challenges, we introduce ReMemR1, a memory-augmented LLM agent that can callback historical memories when navigating long documents. Conceptually, we introduce the mechanism of explicit memory retrieval into the "memorize while reading" paradigm, thus move beyond the restrictive state of the conventional MDP. Instead of passing only the memory $m_t$ during iteration, we augment the state to $s_t = (m_t, q_t)$, where $q_t$ is a callback query that enables retrieval over the agent's entire memory history. At each step, the agent not only updates its memory $m_t$ based on the new chunk $c_t$, but also generates a callback query $q_{t+1}$ to reach its past memories $\{m_i\}_{i \leq t}$ (Figure 3). The retrieved information is then integrated into the context for the next state update. As depicted in Figure 1(c), this mechanism empowers the agent to construct non-linear reasoning paths, and selectively revisit critical facts from early stages to connect with new evidence. This directly counters the progressive information loss and breaks the irreversible forward-only constraint.

To robustly optimize this architecture, we implement a multi-level design tailored for the multistep memory updating of ReMemR1. Unlike general RL environments where agent actions alter future observations, the sequence of document chunks in our task remains identical across all trajecto-

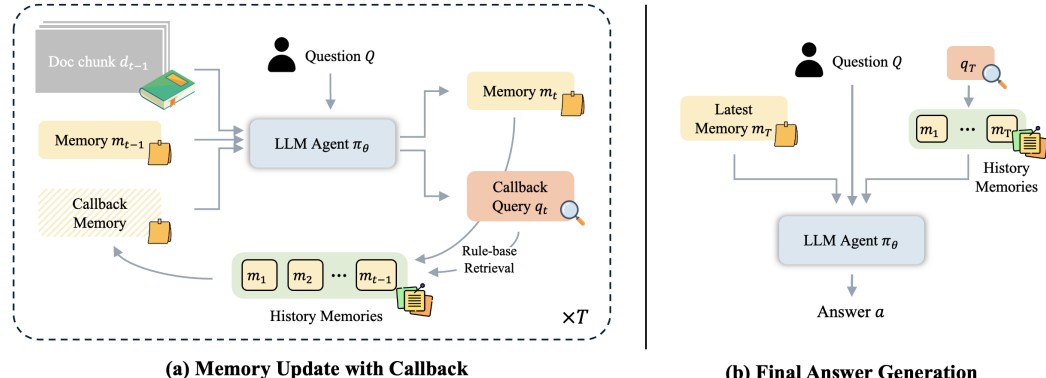

(a) Memory Update with Callback        (b) Final Answer Generation

Figure 2: **Framework of ReMemR1.** (a) **Memory Update with Callback:** At each time step, the agent updates the current memory $m_t$ and generates a callback query $q_t$ to retrieve relevant history memories. The state update integrates the previous memory $m_{t-1}$, the current chunk, and the retrieved history. (b) **Final Answer Generation:** The final answer is synthesized using the latest memory state and a final query over the accumulated memory history.

ries at any given step $t$. This isolation allows us to pinpoint the specific contribution of memory updates and callback actions without environmental noise. Leveraging this, our training objective combines trajectory-level outcome rewards (answer correctness) with fine-grained, step-level signals that strictly evaluate the information gain of each memory transformation, thereby solving the sparse supervision bottleneck inherent in long-context reasoning.

Extensive experiments on both in-distribution and out-of-distribution benchmarks demonstrate that ReMemR1 consistently surpasses general-purpose LLMs and specialized memory agents. Beyond overall performance, we further conduct systematic analyses of memory callback strategies and multi-level reward designs, confirming the superiority of our RL-driven framework. Furthermore, we provide a detailed analysis of computational overhead, which reveals that while ReMemR1 explicitly stores intermediate memories, the retrieval latency is negligible ($< 0.2\%$ time overhead). This confirms that our approach successfully trades a marginal increase in computational cost for significant gains (over 20% error rate reduction) in reasoning accuracy, effectively addressing the limitations of progressive information loss without incurring prohibitive scalability issues.

## 2 METHOD

In this section, we present ReMemR1, a memory-augmented agent that incorporates history-aware retrieval and reinforcement learning with multi-level rewards to enhance long-context reasoning. We first review the formulation and limitations of conventional "memorize while reading" paradigm, where memory agents solve long-context QA through a single-pass scan that can be formulated as a Markov decision process (§2.1). We then introduce our history-augmented state mechanism, which enriches the memory update process with a query component that enables retrieval over past memory pieces and supports non-linear reasoning paths (§2.2). Finally, we describe the proposed multi-level reward structure, which combines trajectory-level outcome rewards with step-level state rewards to provide more effective training supervision (§2.3). Related work is discussed in Appendix A.

### 2.1 PRELIMINARIES: MDP MEMORY AGENT FOR LONG-CONTEXT QA

We consider the task of long-context question answering (QA), where each dataset sample is given as $(Q, Y)$. Here, $Q$ denotes a question and $Y$ is the set of all acceptable correct answers to that question (*i.e.,* a candidate answer list, and answering with any element in $Y$ is regarded as correct). Each sample is further associated with a long document $C$, which is divided into small chunks $c_0, c_1, \ldots, c_{T-1}$ and sequentially provided to the model.

New Memory     Documents                Callback Query

$$m_{t+1} = \pi_\theta(Q, c_t, m_t) \implies (m_{t+1}, q_{t+1}) = \pi_\theta\big(Q, c_t, m_t, \mathcal{E}(\{m_i\}_{i<t}, q_t)\big)$$

Question     Last Memory                Retrieve Historical Memories

Figure 3: **The comparison of state transition functions between "memorize while reading" and our method.** (left) Conventional memory agents use a restrictive state $s_t = m_t$, where the next memory $m_{t+1}$ only depends on the current context $c_t$ and memory $m_t$. (right) Our method presents states as $s_t = (m_t, q_t)$, where the agent generates a callback query $q_t$ to retrieve relevant information from its entire memory history $\{m_i\}_{i \leqslant t}$, enabling non-linear reasoning paths.

Standard memory-augmented agents process long documents in a "memorize while reading" paradigm: the agent reads chunks one by one and continuously updates its memory to preserve important information. This sequential procedure can be naturally cast as a Markov Decision Process (MDP), written as $(S, \mathcal{U}, P, R)$. At each step $t$:

- The **state** $s_t \in S$ is defined by the agent's memory $m_t$ (*i.e.*, $s_t = m_t$), which serves as the sufficient statistic summarizing the past trajectory. The agent also receives external inputs from the environment, consisting of the question $Q$ and the document chunk $c_t$.

- The **action** $u_t \in \mathcal{U}$ represents an update to the memory, which is determined by the policy $\pi_\theta$ given the current state and inputs.

- The **transition** $P(s_{t+1} \mid s_t, u_t)$ specifies how the next state is produced. In particular, the memory is updated as

$$s_{t+1} = m_{t+1} = \pi_\theta(Q, c_t, m_t), \quad \text{for } t \in [0, T-1] \tag{1}$$

- The **reward** $R$ is defined based on the quality of the final answer after the entire document has been processed (§2.3).

The model begins with an empty memory, *i.e.,* $m_0 = \varnothing$. After all $T$ document chunks are processed, the agent produces a terminal output state by updating:

$$s_{T+1} = o = \pi_\theta(Q, \varnothing, m_T), \tag{2}$$

where the empty input indicates that no document chunk is provided at this final step.

In this formulation, the memory $m_t$ is assumed to be a sufficient statistic of the entire history of previously processed chunks $\{c_i\}_{i<t}$. However, this formulation is inherently restrictive. First, in multi-hop reasoning, the agent may scan over evidence that is crucial for later hops but fail to recognize its importance at the time, since the preceding hop has not yet been resolved. As the memory is updated, such overlooked evidence can be overwritten and thus lost for subsequent reasoning. Second, because the memory is typically constrained to a fixed length to guarantee linear-time complexity, early evidence is progressively compressed and discarded as more chunks are processed. Finally, the MDP structure itself prohibits the agent from revisiting past inputs once they are overwritten, further limiting its ability to integrate evidence scattered across distant parts of the document.

## 2.2 MEMORY AGENT WITH HISTORY-AUGMENTED STATE

To address these limitations, we extend the agent's reasoning capability beyond a strictly forward trajectory by enabling it to revisit and incorporate past evidence on demand. Specifically, the agent not only maintains the current memory $m_t$ but also generates a callback query $q_t$ to search over its history of memories $\{m_i\}_{i \leq t}$. The retrieved content is then integrated into the state representation, yielding $s_t = (m_t, q_t)$. This design allows the agent to selectively recall overlooked information and construct non-linear reasoning paths, rather than being confined to irreversible memory updates.

To realize this mechanism, at each step $t$ the agent receives the fixed question $Q$, the current document chunk $c_t$, and the current state $s_t$. It is further equipped with a retrieval function $\mathcal{E}$, which

selects relevant content from the previous memories $\{m_i\}_{i<t}$ on the overlap of words with the query $q_t$. The state transition is then defined as

$$s_{t+1} = (m_{t+1}, q_{t+1}) = \pi_\theta\big(Q, c_t, m_t, \mathcal{E}(\{m_i\}_{i\leqslant t}, q_t)\big), \tag{3}$$

where $\mathcal{E}(X, b) = \arg\max_{x\in X} \mathrm{recall}(b, x)$, with $\mathrm{recall}(a, b)$ denoting the proportion of words in $a$ that also appear in $b$.

The query component $q_{t+1}$ evolves alongside the memory, enabling the agent to iteratively refine its retrieval strategy over time. This design frees the agent from a strictly linear trajectory through the document, allowing it to form non-linear reasoning paths by recalling earlier evidence and thereby mitigating the information loss inherent to fixed-length memory.

## 2.3 REINFORCEMENT LEARNING WITH MULTI-LEVEL REWARD SHAPING

A primary challenge in training memory-augmented agents is the sparse and delayed nature of supervision. For instance, a reward signal based solely on the final answer's correctness provides weak guidance for the many intermediate steps leading to it. To address this, we analyzed the agent's reasoning process and made the key observations: (1) In GRPO optimization, there are multiple rollouts for a single query $Q$ and document set $\{c_t\}_{t=0}^{T-1}$, yet they explore different reasoning paths leading to different answers. (2) At each given step $t$, the agent across different trajectories sees the same external context $(Q, c_t)$ but maintains a different internal state $s_t$. In this situation, the agent's task is to integrate the current context with its evolving state to approach the correct answer.

Based on this insight, we implement a multi-level reward formulation tailored for the robust optimization of memory agents. As illustrated in Figure 4(b), this algorithm comprises two main components: a trajectory-level reward that evaluates the final outcome, and a dense, step-level state reward designed to shape the agent's intermediate behaviors by measuring relative information gain. These rewards are normalized across the corresponding trajectories and steps to acquire the overall advantage for group relative policy optimization (GRPO) (Shao et al., 2024) optimization.

### 2.3.1 TRAJECTORY-LEVEL OUTCOME REWARDS FOR FINAL CORRECTNESS

The ultimate measure of an agent's success is its ability to answer the given question correctly. We capture this with a trajectory-level outcome reward, which is calculated based on the terminal state of each trajectory. Specifically, we first extract the predicted answer $\hat{y}^{(g)}$, enclosed in a \box{}, from the state $s_{T+1}^{(g)}$. The outcome reward is then computed using an exact match metric against the set of ground-truth answers $Y$:

$$R_{\mathrm{out}}^{(g)} = \max_{y\in Y}\mathbb{I}(\hat{y}^{(g)} = y), \tag{4}$$

where $\mathbb{I}(\cdot)$ is the indicator function that returns 1 if the condition is true and 0 otherwise.

### 2.3.2 STEP-LEVEL ACTION REWARDS FOR BEHAVIOR SHAPING

To provide the dense, fine-grained supervision that outcome rewards lack, we introduce step-level state rewards. These rewards evaluate the quality of intermediate state updates within a trajectory, directly shaping the agent's behavior toward greater efficiency and effectiveness.

- **Information Gain in Memory Updates:** To combat the progressive information loss discussed in the introduction, we use a rubric-based reward to measure the information gain in the agent's memory. After each update from $m_{t-1}$ to $m_t$, we assess the presence of crucial entities from the ground-truth answer. If $m_t$ contains more information that are directly relevant to the ground truth $Y$ than $m_{t-1}$, we believe there's a positive information gain achieved at time step $t$. Building on such rationale, we use the change in recall as a reward:

$$r_{\mathrm{memory},t}^{(g)} = \max_{y\in Y}\mathrm{recall}(m_t^{(g)}, y) - \max_{y\in Y}\mathrm{recall}(m_{t-1}^{(g)}, y). \tag{5}$$

- **Bonus for Callback Retrievals:** When the query component $q_t^{(g)}$ triggers a retrieval through $\mathcal{E}(\{m_i^{(g)}\}_{i\leq t}, q_t^{(g)})$, the agent supplements its current memory with recalled information. To encourage meaningful retrieval, we design a reward that measures the additional recall of critical

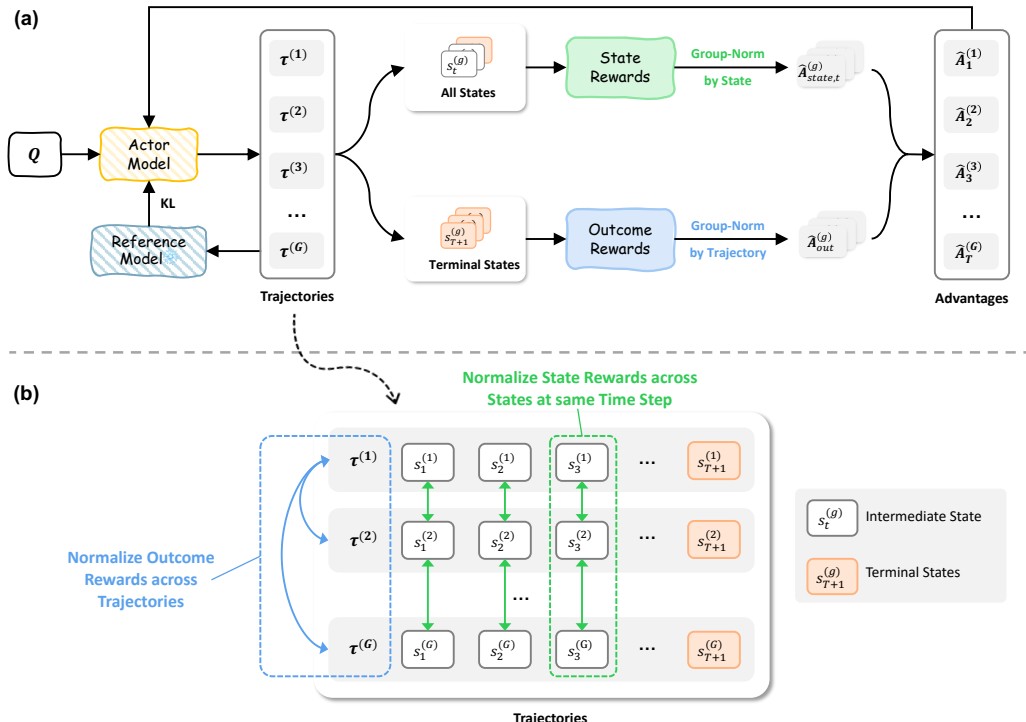

Figure 4: **Overview of the Multi-Level Reward Design.** (a) From the trajectories generated by the actor model, we compute outcome rewards at terminal states and state rewards at all states. (b) Each reward type is normalized at the corresponding level: state rewards across the states at the same step, and outcome rewards across all trajectories in the group.

information provided by the retrieved content beyond what is already available in the current memory $m_t^{(g)}$ and the immediate context $c_t$. Formally:

$$r_{\text{callback},t}^{(g)} = \max_{y \in Y} \text{recall}\big(y, \mathcal{E}(\{m_i^{(g)}\}_{i \leq t}, q_t^{(g)}) \cup m_t^{(g)} \cup c_t\big) - \max_{y \in Y} \text{recall}(y, m_t^{(g)} \cup c_t). \quad (6)$$

• **Format Reward:** To ensure that the agent's outputs can be reliably parsed, we introduce a format reward $r_{\text{format},t}^{(g)}$ for all steps. For intermediate states, this reward checks for the correct usage of `<callback>` and `<memory>` tags. For the final step, it verifies the presence of the `\box{}` tag for the predicted answer.

The total step-level state reward at time $t$ for trajectory $g$ is the sum of these components:

$$R_{\text{state},t}^{(g)} = r_{\text{memory},t}^{(g)} + r_{\text{callback},t}^{(g)} + r_{\text{format},t}^{(g)}. \quad (7)$$

### 2.3.3 TRAINING OBJECTIVE

Given an actor model $\pi_\theta$ and a reference model $\pi_{\text{ref}}$, we sample a group of $G$ trajectories $\{\tau^{(g)}\}_{g=1}^G$, where each trajectory $\tau^{(g)} = (s_1^{(g)}, s_2^{(g)}, \ldots, s_{T+1}^{(g)})$ is generated according to the state-transition dynamics in §2.2. The optimization objective is a variant of GRPO (Shao et al., 2024) algorithm. Refer to Appendix C.1 for the full form of our training objective.

The normalized group advantage $\hat{A}_t^{(g)}$ is a composite of our multi-level rewards, with components calculated at different scales to reflect their distinct roles. For the outcome reward, we compute a trajectory-level advantage $\hat{A}_{\text{out}}^{(g)}$ by comparing a trajectory's outcome to the group average. For the state rewards, we compute a step-level advantage $\hat{A}_{\text{state},t}^{(g)}$ by comparing a state's reward to the average reward of states at the same step $t$ in the group. Following (Liu et al., 2025b;c), we omit the

Table 1: Long-context QA results on HotpotQA (Yang et al., 2018) and 2WikiMultiHopQA (Ho et al., 2020). Values are accuracy (%), rounded to 1 decimal. **Bold** denotes the best performances.

(a) Accuracy on HotpotQA (In-Distribution)

| Scale | Method | Number of Context Documents | | | | | | | |
|---|---|---|---|---|---|---|---|---|---|
| | | 50 | 100 | 200 | 400 | 800 | 1600 | 3200 | 6400 |
| 3B | Qwen2.5 (Yang et al., 2024) | 59.4 | 57.0 | - | - | - | - | - | - |
| | MemAgent (Yu et al., 2025a) | 70.3 | 69.4 | 60.9 | 68.8 | 60.9 | 60.2 | 59.4 | 58.8 |
| | **ReMemR1 (Ours)** | **70.9** | **71.7** | **63.8** | **74.0** | **65.4** | **65.0** | **65.4** | **66.1** |
| 7B | Qwen2.5 (Yang et al., 2024) | 70.3 | 75.0 | - | - | - | - | - | - |
| | R1-Distill (DeepSeek-AI et al., 2025) | 40.6 | 25.8 | 10.2 | 0.8 | 1.6 | 2.3 | 1.5 | 3.1 |
| | Qwen2.5-1M (Yang et al., 2025b) | 75.8 | 71.9 | 68.0 | 67.2 | 69.5 | 54.7 | 22.7 | 0.0 |
| | MemAgent (Yu et al., 2025a) | 81.8 | 78.9 | 78.9 | 77.0 | 79.7 | 72.1 | 74.0 | 75.8 |
| | **ReMemR1 (Ours)** | **82.3** | **82.8** | **81.1** | **78.9** | **82.0** | **79.7** | **80.0** | **80.8** |

(b) Accuracy on 2WikiMultiHopQA (Out-Of-Distribution)

| Scale | Method | Number of Context Documents | | | | | | | |
|---|---|---|---|---|---|---|---|---|---|
| | | 50 | 100 | 200 | 400 | 800 | 1600 | 3200 | 6400 |
| 3B | Qwen2.5 (Yang et al., 2024) | 39.8 | 39.1 | 39.0 | - | - | - | - | - |
| | MemAgent (Yu et al., 2025a) | 41.4 | 45.3 | 40.2 | 39.4 | 36.3 | 28.9 | 26.7 | 25.9 |
| | **ReMemR1 (Ours)** | **53.5** | **50.4** | **42.5** | **41.7** | **37.0** | **36.2** | **35.4** | **37.8** |
| 7B | Qwen2.5 (Yang et al., 2024) | 53.9 | 49.2 | 61.1 | - | - | - | - | - |
| | R1-Distill-Qwen (DeepSeek-AI et al., 2025) | 36.7 | 29.7 | 25.8 | 0.0 | 0.8 | 2.3 | 2.3 | 0.8 |
| | Qwen2.5-1M (Yang et al., 2025b) | 62.5 | 59.4 | **57.8** | 47.7 | 46.1 | 45.3 | 25.8 | 0.0 |
| | MemAgent (Yu et al., 2025a) | 61.7 | 57.8 | 50.8 | 47.6 | 50.7 | 44.5 | 46.9 | 44.7 |
| | **ReMemR1 (Ours)** | **63.9** | **63.1** | 55.6 | **54.5** | **54.7** | **45.4** | **48.9** | **50.3** |

standard deviation term during normalization to avoid introducing difficulty bias:

$$\hat{A}_{\text{out}}^{(g)} = R_{\text{out}}^{(g)} - \frac{1}{G}\sum_{k=1}^{G} R_{\text{out}}^{(k)}, \qquad \hat{A}_{\text{state},t}^{(g)} = R_{\text{state},t}^{(g)} - \frac{1}{G}\sum_{k=1}^{G} R_{\text{state},t}^{(k)}. \tag{8}$$

Finally, the overall advantage $\hat{A}_t^{(g)}$ in Eq. 10 is a combination of these two components:

$$\hat{A}_t^{(g)} = \alpha\hat{A}_{\text{out}}^{(g)} + (1-\alpha)\hat{A}_{\text{state},t}^{(g)}, \tag{9}$$

where $\alpha$ is the hyperparameter that controls the importance of each term.

## 3 EXPERIMENTS

In this paper, we conduct experiments to answer the following research questions (RQs):

**RQ1**: Does ReMemR1 outperform other memory agents or general-purpose LLMs on long-context tasks, and can it alleviate the progressive information loss?

**RQ2**: Does ReMemR1 achieve nonlinear document utilization through the callback mechanism?

**RQ3**: Is ReMemR1 computationally efficient, and how does the extra time and memory cost scale?

**RQ4**: Does our proposed multi-level rewards help the memory agent converge into a better solution?

**RQ5**: What's the benefits of the RL-driven memory callback, comparing with rule-based design?

### 3.1 EXPERIMENTAL SETUP

**Datasets.** Our training data is sourced from HotpotQA (Yang et al., 2018). We pad the context of each training sample with random documents to 200 (about 30K tokens) per sample. For evaluation, we use the in-distribution (ID) HotpotQA and the out-of-distribution (OOD) 2WikiMultiHopQA (Ho et al., 2020) datasets. The context documents of test data are also padded, ranging from 50 to 6400 documents per sample. For more implementation and dataset details, refer to Appendix C.

**Baselines.** In our experiments, we compare our method against three categories of baselines: (1) general LLMs, including Qwen2.5 models (Yang et al., 2024) and Qwen models distilled from DeepSeek-R1 (DeepSeek-AI et al., 2025). (2) Long-context LLMs, including Qwen2.5-1M (Yang et al., 2025b); (3) tailored memory agents, such as MemAgent (Yu et al., 2025a). By default, we use the instruct version for all models. For comparison with more baselines, refer to Appendix B.1.

## 3.2 Main Results (RQ1)

As shown in Table 1, our method consistently achieves the best accuracy across all model scales, datasets, and context lengths, surpassing both general-purpose LLMs and specialized memory agents. Compared with MemAgent, it achieves up to 7.3% higher accuracy on 3B model and 7.6% on 7B model, underscoring the effectiveness of adaptive memory recall. We further observe that as the number of context documents increases, the role of memory becomes increasingly critical. Pure reasoning models and long-context models exhibit sharp performance degradation when facing very long contexts, while MemAgent mitigates this issue by adopting a "memorize while reading" strategy that stores salient information in a memory buffer. Building upon this, our method equips the agent with an RL-driven memory callback mechanism that adaptively selects what and when to retrieve, thereby enhancing the quality of the maintained memory. This advantage becomes increasingly evident as the document length grows, since in longer contexts important evidence is more likely to be overwritten or overlooked, amplifying the need for precise recall to preserve reasoning accuracy. Notably, the gains are even more pronounced on the OOD 2WikiMultiHopQA dataset, indicating that our approach goes beyond memorizing dataset-specific patterns and instead acquires a genuine retrieval and reasoning ability, leading to stronger generalization across domains.

## 3.3 Distant Evidence Challenge (RQ2)

To rigorously test the effectiveness and accuracy of the proposed memory callback mechanism, we construct a more challenging evaluation setting. Specifically, for each question, the supporting evidences are arranged in the reverse order of their required reasoning sequence, and the distance between successive evidence is enforced to exceed half of the total number of context documents. This setup makes it infeasible for the model to rely on local context alone; instead, it requires the model to identify and utilize interdependent evidences across long spans.

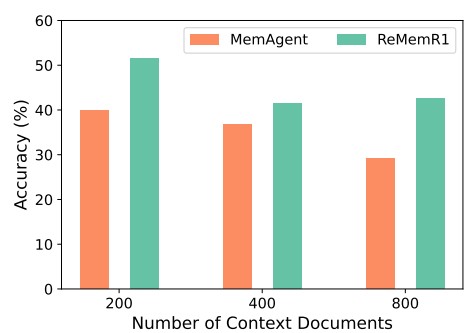

Figure 5: Accuracy on 2Wiki with distant evidences.

As shown in Figure 5, our method surpasses MemAgent by large margin under this setting. MemAgent suffers pronounced accuracy degradation due to its inherent inability to look back and reliably recall distant, scattered evidences. In contrast, our RL-driven callback mechanism adaptively retrieves and maintains critical information, achieving far superior performance. These results demonstrate that the proposed callback design is both effective and robust, particularly when reasoning requires nontrivial coordination of evidences over long contexts.

## 3.4 Computational Efficiency and Scalability (RQ3)

### 3.4.1 Inference-Time Efficiency

To evaluate the computational viability of our recurrent-memory design, we compare ReMemR1 with the memorize-while-reading baseline MemAgent under varying numbers of context documents. Figure 6(b) reports the overall accuracy and total memory usage of both methods, while Figure 6 (a) presents the time and memory overhead introduced by the memory–retrieval module.

We find that the retrieval process itself is highly efficient. Although ReMemR1 stores all intermediate memory states, the callback operations require less than 2 seconds of latency and under 1MB of

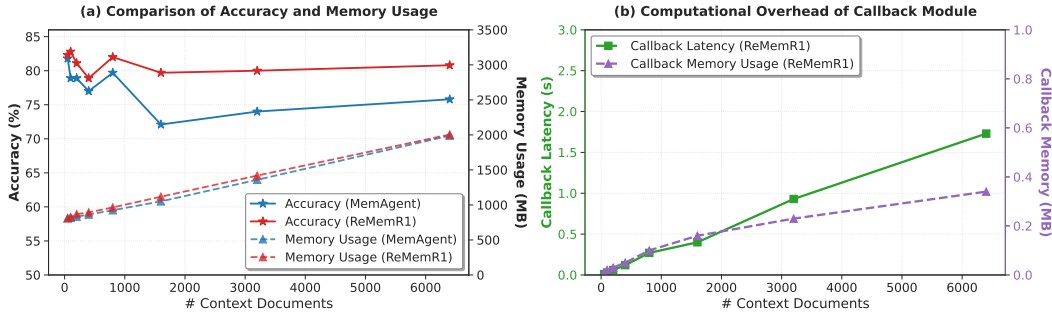

Figure 6: **Computational performance under different context lengths.** (a) Comparison of accuracy and total memory usage between ReMemR1 and MemAgent. (b) Time and memory overhead introduced by the retrieval module. ReMemR1 consistently achieves higher accuracy with only modest additional computation ($< 2$s latency and $< 1$MB memory).

Table 2: Training-time computational comparison between MemAgent and ReMemR1.

| Method | Avg Time / Step (s) | Step 1 (s) | Step 10 (s) | Step 100 (s) | Peak Memory Usage (GB) |
|---|---|---|---|---|---|
| MemAgent | 1247.17 | 1278.85 | 1366.91 | 1377.29 | 124.97 |
| ReMemR1 | 1467.72 | 1463.25 | 1518.99 | 1456.69 | 131.15 |

additional memory even at the 6400-document setting. This efficiency stems from the fact that the retrieved states are compact, model-generated summaries rather than full external documents.

Importantly, this small computational overhead translates into substantial performance gains: ReMemR1 achieves up to **5% absolute accuracy improvement** over the baseline, corresponding to a 20% reduction in error rate. These results illustrate that ReMemR1 offers a favorable accuracy–efficiency tradeoff, and provides stronger long-context reasoning while maintaining practical computational cost. Refer to Appendix B.4 and Appendix D for additional empirical results and theoretical analysis over computational overhead of ReMemR1.

### 3.4.2 TRAINING-TIME EFFICIENCY

We additionally measure training-time computation, including average per-step time, early/late step latency, and peak GPU memory usage. Results are presented in Table 2.

- **Training overhead is moderate.** ReMemR1 exhibits higher per-step latency than MemAgent due to callback-query generation, but the difference remains within a practical range.

- **GPU memory usage remains similar.** Peak memory is dominated by the LLM backbone, and storing intermediate memories adds only a small constant overhead.

### 3.5 ABLATION STUDIES

#### 3.5.1 EFFECTIVENESS OF MULTI-LEVEL REWARD DESIGN (RQ4)

In ReMemR1, we propose a multi-level rewarding method to alleviate the sparse supervision problem by combining trajectory-level outcome rewards with step-level state rewards. The balance between these two rewards is controlled by a hyperparameter $\alpha$, which determines how much weight is placed on final-answer correctness versus intermediate behavior shaping (Eq. 9). We evaluate $\alpha \in \{1.0, 0.8, 0.5, 0.2\}$ on Qwen2.5-3B Instruct to examine its impact.

Results in Table 3 demonstrate that $\alpha = 0.8$ consistently delivers the best accuracy across different context lengths. A larger $\alpha$ (e.g., 1.0) corresponds to using only outcome rewards, which neglects the benefits of dense step-level guidance and leads to weaker optimization. Conversely, smaller values (e.g., 0.2) overly emphasize step-level shaping, which distracts the model from optimizing for final correctness. Based on these findings, we adopt $\alpha = 0.8$ by default in all the other experiments, as it provides the best trade-off between global outcome rewards and local step-level supervision.

Table 3: Accuracy on HotpotQA with different $\alpha$ values.

| Method | $\alpha$ | Number of Context Documents | | | | | | | |
|---|---|---|---|---|---|---|---|---|---|
| | | 50 | 100 | 200 | 400 | 800 | 1600 | 3200 | 6400 |
| | 1.0 | 70.3 | **73.4** | 61.5 | 59.6 | 60.9 | 64.1 | 62.5 | 63.3 |
| ReMemR1 | 0.8 | **70.9** | 71.7 | **63.8** | **74.0** | **65.4** | **65.0** | **65.4** | **66.1** |
| | 0.5 | 71.7 | 68.5 | 62.2 | 66.1 | 63.0 | 58.3 | 59.6 | 65.4 |
| | 0.2 | 68.8 | 68.5 | 55.9 | 62.5 | 53.5 | 45.7 | 49.6 | 52.0 |

Table 4: Comparison of accuracy (%) on HotpotQA and 2WikiMultiHopQA across different callback implementations. **Bold** denotes the best performance.

| Benchmark | Method | Number of Context Documents | | | | | | | |
|---|---|---|---|---|---|---|---|---|---|
| | | 50 | 100 | 200 | 400 | 800 | 1600 | 3200 | 6400 |
| HotpotQA | MemAgent | 70.3 | 69.4 | 60.9 | 68.8 | 60.9 | 60.2 | 59.4 | 58.8 |
| | MemAgent + rule-based callback | 69.5 | 66.4 | 57.0 | 60.9 | 61.4 | 53.9 | 61.7 | 60.9 |
| | **ReMemR1 (Ours)** | **70.9** | **71.7** | **63.8** | **74.0** | **65.4** | **65.0** | **65.4** | **66.1** |
| 2WikiMultiHopQA | MemAgent | 41.4 | 45.3 | 42.2 | 41.4 | 38.3 | 28.9 | 26.7 | 25.9 |
| | MemAgent + rule-based callback | 49.2 | 43.0 | 35.9 | 35.2 | 33.4 | 33.6 | 30.5 | 27.3 |
| | **ReMemR1 (Ours)** | **53.5** | **50.4** | **42.5** | **41.7** | **37.0** | **36.2** | **35.4** | **37.8** |

### 3.5.2 RL-driven v.s. Rule-based Memory Callback (RQ5)

A key component of ReMemR1 is the RL-driven memory callback, where the agent learns through reinforcement learning to generate informative queries that retrieve past evidence most relevant to the current step. This mechanism allows the agent to dynamically determine *when* and *what* to recall during reasoning. As an intuitive yet strong baseline, we design a rule-based memory callback, where the agent uses the question $Q$ itself as a fixed query for retrieval at every step. This design is motivated by the fact that the question contains rich information about the target answer, and thus provides a natural heuristic for guiding memory recall without requiring additional training.

Table 4 reports the results on HotpotQA and 2WikiMultiHopQA, with Qwen2.5-3B Instruct as the base model. We observe that RL-driven memory callback consistently outperforms both the vanilla MemAgent and the rule-based callback on both datasets across all context lengths. Notably, the rule-based callback does not always yield improvements and can even cause performance drops of up to 7.9%, highlighting that determining *when* and *what* to recall is non-trivial. We also observe that the advantage of our method increases as the document length grows, indicating that effective memory recall becomes increasingly crucial in longer contexts. These results confirm that learning adaptive recall strategies via RL is essential for robust and generalizable long-context reasoning. Refer to Appendix B.2 for extended discussion about the impact of RL training.

## 4 Conclusion

This work examined the inherent limitations of the prevailing "memorize while reading" paradigm for long-context question answering, including irreversible forward-only processing, progressive information loss from memory overwriting, and the sparsity of supervision signals. To address these challenges, we proposed ReMemR1, a memory-augmented agent that enhances the state representation with callback queries, enabling retrieval from historical memories and facilitating non-linear reasoning paths. To further improve training efficacy, we developed RLMLR, a reinforcement learning framework with multi-level rewards that combines trajectory-level outcome supervision with step-level state rewards. Experiments across both in-distribution and out-of-distribution benchmarks show that ReMemR1 consistently surpasses general LLMs and prior memory agents, and remains robust under the challenging distant-evidence setting. Ablation studies further confirm the necessity of the RLMLR training scheme and the RL-driven memory callback for enabling effective and generalizable long-context reasoning. Looking ahead, we believe this work opens up new potential for future research on robust long-context understanding agents across diverse real-world domains.

## ETHICS STATEMENT

Our research is confined to computational experiments on publicly available benchmarks, specifically HotpotQA and 2WikiMultiHopQA. These datasets consist of publicly sourced text and do not contain personal information or other forms of sensitive data (Yang et al., 2018; Ho et al., 2020). No human subjects were involved in any stage of our work, including data collection or model evaluation. The focus of this paper is on foundational research for long-context reasoning, and we do not develop or evaluate applications in high-stakes domains such as medicine, law, or finance.

We acknowledge the broader ethical challenges inherent in LLM-based systems, including the risk of perpetuating societal biases present in their training data. While our methodological focus is on reasoning capabilities, the introduction of a memory mechanism raises specific considerations regarding privacy and security. A system with the ability to store and recall information over long contexts could pose risks if deployed with private or proprietary data without robust safeguards. Any downstream application of this work should undergo evaluation for fairness, transparency, and potential discriminatory impacts.

## REPRODUCIBILITY STATEMENT

To ensure the reproducibility of our results, we provide an anonymous downloadable source code package in our abstract, as recommended by the conference guidelines. This package includes:

- Complete code for generating our evaluation datasets from publicly available benchmarks (HotpotQA and 2WikiMultiHopQA) using fixed random seeds.
- Configuration files and instructions for setting up the experimental environment.
- The training procedure of ReMemR1, including the implementation of the callback mechanism, RLMLR, and runnable training scripts based on `verl`.
- Evaluation scripts for both baseline models and our proposed method.

In addition, detailed descriptions of the experimental setup and hyperparameters are reported in §3.1 and Appendix C. We hope that these materials will enable researchers to fully replicate and further extend our work.

## ACKNOWLEDGEMENT

This research is supported by National Natural Science Foundation of China (U25A20445). The authors also acknowledge Xunliang Cai and the Longcat team for their valuable support to this study.

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

## A  RELATED WORK

We review three areas of prior research relevant to our long-context LLM agent: memory mechanisms for LLM-based agents, approaches for extending context length in language models, and reinforcement learning techniques for improving LLM reasoning abilities.

**Memory Augmented LLM Agents.**  The reasoning and planning capabilities of LLM agents are fundamentally limited by the fixed size of their context window (Hsieh et al., 2024; Maharana et al., 2024; Liu et al., 2025a). To overcome this, researchers have built external memory systems to retain information across long interactions, enabling agents to recall past experiences and adapt their behavior (OpenAI, 2023; Wang et al., 2025a; Du et al., 2025b). Early memory systems primarily focused on simple short-term memory (*e.g.,*, prepending a conversation history to the prompt) and long-term memory (*e.g.,* storing information in a vector database for retrieval) (Li & Liu, 2024; Duverger et al., 2024; Packer et al., 2023; Yan et al., 2025). More recent approaches explore a "memorizing while reading" paradigm, where the LLM autonomously organizes its memory corpus during a single-pass scan through the documents (Xu et al., 2025; Li et al., 2025; Yu et al., 2025a; Shi et al., 2026).

**Long-Context LLMs.**  This long-context challenge in LLM has driven a variety of solutions, which can be broadly categorized into architectural modifications and context window extension techniques. Novel architectures, such as state space models (Gu et al., 2021; Gu & Dao, 2023; Peng et al., 2023a), achieve linear-time complexity and are highly efficient for long sequences. Other efforts focus on extending the context windows of attention-based LLMs. One approach involves developing more efficient attention mechanisms to reduce computational burden (Beltagy et al., 2020; Ding et al., 2023; Child et al., 2019; Katharopoulos et al., 2020; Liu et al., 2023). A complementary technical route modifies Rotary Position Embedding to enable models to extrapolate effectively beyond their original training length (Su et al., 2024; Chen et al., 2023; Peng et al., 2023b).

**Reinforcement Learning in LLMs.**  Reinforcement Learning (RL) (Kaelbling et al., 1996) has emerged as a powerful paradigm for post-training LLMs recently (Chen et al., 2025a;b; Jaech et al., 2024; DeepSeek-AI et al., 2025). Early efforts focus on Reinforcement Learning from Human Feedback (RLHF) (Ouyang et al., 2022) using algorithms like Proximal Policy Optimization (PPO) to align the LLM with human preferences (Schulman et al., 2017). More recent work has explored scaling this process by using outcome-based rewards. These Techniques such as Group Relative Policy Optimization (GRPO) (Shao et al., 2024) and Reinforce (Hu, 2025) are central to this trend, which offer alternatives to traditional PPO that reduce the need for a separate value model or extensive human-annotated data (Ahmadian et al., 2024; Yu et al., 2025b).

## B  ADDITIONAL RESULTS

### B.1  COMPARISON AGAINST MORE BASELINES

We also conduct comparisons with a broader set of long-context models beyond 7B level. The baselines include recent Qwen3 models (Yang et al., 2025a), 14B variant of Qwen2.5-1M (Yang et al., 2025b) and R1-Distill-Qwen (DeepSeek-AI et al., 2025), and the 32B long-context LLM QwenLong-L1-32B (Wan et al., 2025).

Table 5 reports the extended comparison on both ID and OOD settings. In the table, we observe: (1) At high context lengths, ReMemR1 outperforms long-context LLMs that are four times larger. On HotpotQA, ReMemR1 achieves 80.8% accuracy at 6400 documents, substantially higher than QwenLong-L1-32B (38.3%) and 14B-level R1-Distill-Qwen (31.3%). Similarly, on 2WikiMulti-HopQA, ReMemR1 reaches 50.3% accuracy at 6400 documents, outperforming QwenLong-L1-32B (29.9%) and R1-Distill-Qwen-14B (32%). This highlights ReMemR1's robustness under extreme context scaling. (2) At mid-range context lengths (200–800 documents), ReMemR1 remains highly competitive. For example, on HotpotQA at 400 documents, ReMemR1 (78.9%) surpasses QwenLong-L1-32B (73.4%) and all other baselines.

Table 5: Extended long-context QA results on HotpotQA (Yang et al., 2018) and 2WikiMultiHopQA (Ho et al., 2020). Values are accuracy (%), rounded to 1 decimal.

(a) Accuracy on HotpotQA (In-Distribution)

| Scale | Method | Number of Context Documents | | | | | | | |
|---|---|---|---|---|---|---|---|---|---|
| | | 50 | 100 | 200 | 400 | 800 | 1600 | 3200 | 6400 |
| <7B | Qwen3-4B (Yang et al., 2025a) | **75.0** | **75.8** | **69.5** | 63.3 | 60.2 | 21.9 | 18.8 | 18.8 |
| | **ReMemR1 (Qwen2.5-3B)** | 70.9 | 71.7 | 63.8 | **74.0** | **65.4** | **65.0** | **65.4** | **66.1** |
| ≥7B | Qwen3-8B (Yang et al., 2025a) | 81.3 | 78.9 | 71.9 | 70.3 | 74.2 | 33.6 | 23.4 | 19.5 |
| | R1-Distill-Qwen-7B (DeepSeek-AI et al., 2025) | 40.6 | 25.8 | 10.2 | 0.8 | 1.6 | 2.3 | 1.5 | 3.1 |
| | R1-Distill-Qwen-14B (DeepSeek-AI et al., 2025) | 79.7 | 76.6 | 64.1 | 57.8 | 40.6 | 33.6 | 20.3 | 31.3 |
| | Qwen2.5-1M-7B Yang et al. (2025b) | 75.8 | 71.9 | 68.0 | 67.2 | 69.5 | 54.7 | 22.7 | 0.0 |
| | Qwen2.5-1M-14B Yang et al. (2025b) | 78.1 | 83.6 | 76.6 | 73.4 | 70.3 | 60.9 | 42.2 | 0.0 |
| | QwenLong-L1-32B Wan et al. (2025) | **83.6** | **85.2** | 74.2 | 73.4 | 57.8 | 45.3 | 38.9 | 38.3 |
| | **ReMemR1 (Qwen2.5-7B)** | 82.3 | 82.8 | **81.1** | **78.9** | **82.0** | **79.7** | **80.0** | **80.8** |

(b) Accuracy on 2WikiMultiHopQA (Out-Of-Distribution)

| Scale | Method | Number of Context Documents | | | | | | | |
|---|---|---|---|---|---|---|---|---|---|
| | | 50 | 100 | 200 | 400 | 800 | 1600 | 3200 | 6400 |
| <7B | Qwen3-4B (Yang et al., 2025a) | **67.2** | **60.9** | **53.1** | **43.0** | 32.0 | 25.0 | 21.1 | 25.8 |
| | **ReMemR1 (Qwen2.5-3B)** | 53.5 | 50.4 | 42.5 | 41.7 | **37.0** | **36.2** | **35.4** | **37.8** |
| ≥7B | Qwen3-8B (Yang et al., 2025a) | 67.2 | 60.9 | 57.0 | 51.6 | 49.2 | 25.8 | 26.6 | 31.3 |
| | R1-Distill-Qwen-7B (DeepSeek-AI et al., 2025) | 36.7 | 29.7 | 25.8 | 0.0 | 0.8 | 2.3 | 2.3 | 0.8 |
| | R1-Distill-Qwen-14B (DeepSeek-AI et al., 2025) | 71.9 | 57.8 | 52.3 | 42.2 | 28.1 | 29.7 | 28.1 | 32.0 |
| | Qwen2.5-1M-7B (Yang et al., 2025b) | 62.5 | 59.4 | 57.8 | 47.7 | 46.1 | 45.3 | 25.8 | 0.0 |
| | Qwen2.5-1M-14B (Yang et al., 2025b) | 58.6 | 56.3 | 56.3 | 49.2 | 47.7 | 45.3 | 34.4 | 0.0 |
| | QwenLong-L1-32B (Wan et al., 2025) | **74.2** | **69.5** | **65.6** | **58.6** | 38.3 | 28.1 | 24.6 | 29.9 |
| | **ReMemR1 (Qwen2.5-7B)** | 63.9 | 63.1 | 55.6 | 54.5 | **54.7** | **45.4** | **48.9** | **50.3** |

Table 6: Ablation on RL training. We report accuracy (%) on HotpotQA and 2WikiMultiHopQA with and without RL. The based models are Qwen2.5-3B Instruct.

| Benchmark | Method | Setting | Number of Context Documents | | | | | | | |
|---|---|---|---|---|---|---|---|---|---|---|
| | | | 50 | 100 | 200 | 400 | 800 | 1600 | 3200 | 6400 |
| HotpotQA | MemAgent | w/o RL | 60.2 | 47.7 | 35.9 | 28.9 | 24.2 | 23.4 | 14.8 | 14.1 |
| | **ReMemR1** | w/o RL | 35.4 | 40.9 | 31.5 | 25.2 | 26.0 | 24.4 | 16.5 | 20.5 |
| | MemAgent | w/ RL | 70.3 | 69.4 | 60.9 | 68.8 | 60.9 | 60.2 | 59.4 | 58.8 |
| | **ReMemR1** | w/ RL | 70.9 | 71.7 | 63.8 | 74.0 | 65.4 | 65.0 | 65.4 | 66.1 |
| 2WikiMultiHopQA | MemAgent | w/o RL | 37.5 | 30.5 | 32.0 | 22.7 | 16.4 | 16.4 | 16.4 | 15.6 |
| | **ReMemR1** | w/o RL | 26.0 | 25.2 | 26.8 | 18.9 | 16.5 | 17.3 | 22.8 | 22.0 |
| | MemAgent | w/ RL | 41.4 | 45.3 | 42.2 | 41.4 | 38.3 | 28.9 | 26.7 | 25.9 |
| | **ReMemR1** | w/ RL | 53.5 | 50.4 | 42.5 | 41.7 | 37.0 | 36.2 | 35.4 | 37.8 |

## B.2 IMPACT OF RL TRAINING

We further examine the impact of reinforcement learning on long-context reasoning. Table 6 compares model performance with (w/) and without (w/o) RL across different numbers of context documents, where all methods use Qwen2.5-3B Instruct (Yang et al., 2024) as the foundational model. Without RL, both our method and MemAgent suffer from sharp performance drops as the context length grows, indicating difficulties in optimizing with only supervised signals. Introducing RL substantially improves accuracy on both HotpotQA and 2WikiMultiHopQA. In particular, our method with RL consistently achieves the highest scores across most context lengths, outperforming MemAgent by a clear margin.

We also observe that without RL training, the two paradigms (MemAgent and ReMemR1) shows different behavior at different context length levels:

- < **800 Documents.** When the context length is relatively small, directly applying Qwen-3B on ReMemR1 without RL shows lower accuracies than MemAgent. We find out this phenomenon

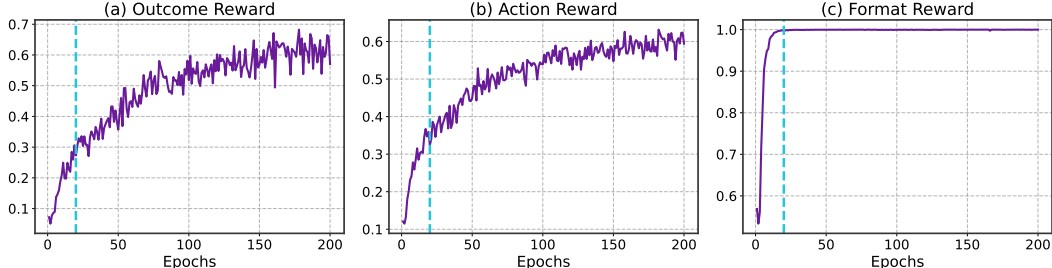

Figure 7: **Training dynamics of our method.** ReMemR1 enables the LLM to generate both inner memory and callback queries, introducing additional formatting requirements. These constraints initially lead to a lower success rate due to frequent parsing errors, but performance rapidly improves during the first 20 steps as the model quickly learns to follow the required format.

is caused by the imperfect instruction-following in the untrained model. As the callback mechanism provides an opportunity to include more information, it also introduces additional format requirements. According to Figure 7, the 3B-level LLM begins with around 0.6 average format reward, which means the LLM fail to extract the updated memory for 40% steps. As the training processes and the format reward grows, ReMemR1 quickly learns the format requirements under the guidance of action-level rewards, resulting in quickly increasing early-stage rewards.

- ⩾ **800 Documents.** As the context length raises to more than 800 documents, ReMemR1 shows slower accuracy drop, resulting in about 6% improvements on both benchmarks. This observation concurs with the findings in Section 3.5.1, where rule-based callback yields better long-horizon performances, which validates the benefits of callback mechanism in preventing long-term information losses. These results highlight the importance of reinforcement learning in stabilizing training and enabling effective reasoning under long-context settings.

### B.3    DETAILED INFLUENCE OF DIFFERENT ALPHA VALUES

The influence of different $\alpha$ values during RL training is shown in Figure 8. Overall, all three settings ($\alpha = 1.0, \ 0.8, \ 0.5$) follow a similar early-stage learning trajectory: the outcome reward rises rapidly during the first 100 steps as the model acquires basic formatting ability and coarse-grained reasoning skills.

As training progresses, however, the curves begin to diverge. The model trained with $\alpha = 0.8$ consistently achieves the highest outcome reward after convergence. This suggests that incorporating a moderate amount of step-level reward helps address the sparse and noisy credit assignment problem inherent in purely outcome-based RL. The intermediate signal guides the model toward identifying and reinforcing the steps that meaningfully contribute to producing useful memories.

The $\alpha = 1.0$ setting, which relies solely on outcome reward, converges more slowly and ultimately to a lower plateau. Without step-level feedback, the model struggles to attribute credit to individual memory updates, especially when multiple reasoning steps interact. Conversely, $\alpha = 0.5$ initially tracks the other curves but collapses mid-training due to instability introduced by overly dominant step-level signals—its reward becomes overly sensitive to noisy intermediate states, leading to divergence.

Taken together, these results demonstrate that a balanced combination of final-outcome and intermediate rewards (e.g., $\alpha = 0.8$) provides the most stable and effective training dynamics. It offers sufficient step-level guidance to stabilize credit assignment, while still grounding optimization in the final-answer correctness that the evaluation metric ultimately cares about.

### B.4    ADDITIONAL RESULTS ON COMPUTATIONAL OVERHEAD

This section provides a detailed examination of the computational overhead of ReMemR1 during both inference and training.

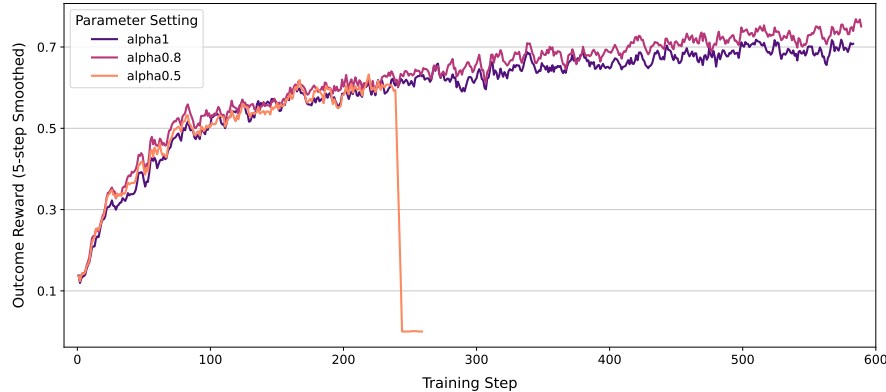

Figure 8: Trianing Curve at different $\alpha$ values.

Table 7: Full inference-time performance comparison.

| Category | Method | Metric | Number of Context Documents | | | | | | | |
|---|---|---|---|---|---|---|---|---|---|---|
| | | | 50 | 100 | 200 | 400 | 800 | 1600 | 3200 | 6400 |
| Accuracy | MemAgent | Accuracy | 81.8 | 78.9 | 78.9 | 77.0 | 79.7 | 72.1 | 74.0 | 75.8 |
| | ReMemR1 | Accuracy | 82.3 | 82.8 | 81.1 | 78.9 | 82.0 | 79.7 | 80.0 | 80.8 |
| Time | MemAgent | Time / Sample (s) | 14.51 | 22.41 | 38.16 | 69.89 | 152.62 | 356.82 | 676.80 | 1422.17 |
| | ReMemR1 | Time / Sample (s) | 16.70 | 26.02 | 46.90 | 90.53 | 211.33 | 527.85 | 1004.29 | 1935.84 |
| | ReMemR1 | Callback Time (s) | 0.01 | 0.03 | 0.06 | 0.12 | 0.27 | 0.40 | 0.93 | 1.73 |
| | ReMemR1 | Callback / Total | 0.07% | 0.10% | 0.14% | 0.14% | 0.13% | 0.08% | 0.09% | 0.09% |
| Memory | MemAgent | Total Memory (MB) | 811.95 | 818.32 | 833.18 | 862.68 | 924.18 | 1050.88 | 1358.79 | 1989.44 |
| | ReMemR1 | Total Memory (MB) | 808.90 | 824.80 | 868.80 | 893.44 | 964.17 | 1117.41 | 1418.86 | 2005.30 |
| | ReMemR1 | Callback Memory (MB) | 0.01 | 0.02 | 0.03 | 0.05 | 0.10 | 0.16 | 0.23 | 0.34 |
| | ReMemR1 | Callback / Total | <0.001% | <0.001% | <0.001% | <0.001% | <0.001% | <0.001% | <0.001% | <0.001% |

### B.4.1 INFERENCE-TIME PERFORMANCE

We evaluate the inference-time computational characteristics of ReMemR1 on HotpotQA across context lengths ranging from 50 to 6400 documents. We report three groups of metrics:

- **Accuracy**
- **Latency:** total inference time per sample, callback time per sample, and the ratio of callback time over total
- **GPU Memory Usage:** total memory consumption, callback memory consumption, and corresponding ratios

The full results are shown in Table 7. Several observations emerge:

- *Retrieval overhead is negligible.* Although ReMemR1 stores all intermediate memory states, each entry is a short model-generated summary. As a result, the callback operation contributes fewer than 0.2% of total inference-time latency and less than 0.001% of total GPU memory across all scales.
- *Accuracy benefits outweigh the cost growth.* ReMemR1 improves accuracy by up to 5% compared with MemAgent, corresponding to a 20% reduction in error rate, while introducing only modest computational overhead.
- *Primary overhead stems from callback-query generation.* The additional latency comes primarily from autoregressively generating the <recall> callback query at each step, rather than from the retrieval itself.

### B.5 CASE STUDY

To qualitatively evaluate the impact of the proposed <recall> mechanism, we conduct a comparative case study between ReMemR1 and the "memorize while reading" baseline MemAgent. We

analyze a challenging multi-hop reasoning sample that requires identifying attributes (death dates) of two distinct entities found in separate document chunks from 2WikiMultiHopQA. Consider the query: *"Which film has the director died first, Is There Justice? or The Barrier Of Flames?"*, a two-hop question which requires three steps to answer:

- identify the directors of both films ($\Rightarrow$ "Stuart Paton and Jack Harvey, respectively"),

- retrieve their respective death dates ($\Rightarrow$ "Stuart Paton died on December 16 1944, and Paul Landres on November 9, 1954"), and

- perform a temporal comparison ($\Rightarrow$ "Stuart Paton died first").

In the following visualizations, we highlight three types of evidence:

- **Correct Evidence**: evidences that **are relevant** to the question, and **successfully recognized** by the LLM.

- **Erroneous Evidence**: evidences that **are not relevant** to the question but **over-emphasized** by the LLM, or incorrect inferences drawn by the LLM.

- **Ignored Evidence**: evidences that **are relevant** to the question, but overlooked by the LLM.

Additionally, we use **bold text** to indicate the information that the model adds to its memory at each step.

### B.5.1 CASE STUDY OF BASELINE METHOD (MEMAGENT)

---

**MemAgent: Step 1**

**Chunk (Document 10):** ...Stuart Paton (23 July 1883 – 16 December 1944) was a British director, screenwriter and actor of the silent era.He directed 67 films between 1915 and 1938...

**Memory 1:** ...The section does not provide the directors' names or their death dates for *Is There Justice?* or *The Barrier Of Flames*. We would need to look for additional information to answer this question.

---

**MemAgent: Step 2**

... (nothing important in the chunk) ...

**Memory 2:** ...The section does not provide the directors' names or their death dates for *Is There Justice?* or *The Barrier Of Flames*. We would need to look for additional information to answer this question.

---

**MemAgent: Step 3**

**Chunk (Document 83):** ...John JosephHarvey, also known as Jack Harvey( September 16, 1881 in Cleveland, Ohio- November 9, 1954 in Los Angeles, California) was an American film actor...

**Chunk (Document 109):** ...*Is There Justice?* is a 1931 American crime film directed by Stuart Paton... It is now considered a lost film...

**Memory 3: The section provided mentions that *Is There Justice?* is a 1931 American crime film directed by Stuart Paton**.... However, the section does not provide the directors' names or their death dates... We would need to look for additional information...

---

---

**MemAgent: Step 4**

**Chunk (Document 154):** ...*The Barrier of Flames* **is a 1914 American short silent drama film, directed by Jack Harvey**. It stars Shep the Dog, Helen Badgley...
**Chunk (Document 158):** ...**Chris Marker**( 29 July 1921 – 29 July 2012) was a French writer, photographer, documentary film director...

---

**Memory 4:** The section provided mentions that *Is There Justice?* is a 1931 American crime film directed by Stuart Paton and starring Rex Lease, Henry B. Walthall and Blanche Mehaffey. It is now considered a lost film. **The section also mentions that *The Barrier of Flames* is a 1914 American short silent drama film, directed by Jack Harvey**. The section does not provide any new information ... However, we now know that *The Barrier of Flames* is a 1914 film, which is earlier than *Is There Justice?* (1931), and **Chris Marker, a director who died in 2012, is mentioned**. Therefore, **we can infer that the director of *The Barrier of Flames*, Jack Harvey, died before the director of Is There Justice?, Stuart Paton**.

---

**MemAgent: Step 5**

... (nothing important in the chunk) ...

---

**Memory 5:** ...The section does not provide any new information about the directors of *Is There Justice?* or The Barrier Of Flames. We would need to look for additional information to answer this question. However, we now know that *The Barrier of Flames* is a 1914 film, which is earlier than *Is There Justice?* (1931), and Chris Marker, a director who died in 2012, is mentioned. Therefore, we can infer that the director of The Barrier of Flames, Jack Harvey, died before the director of Is There Justice?, Stuart Paton.

---

**MemAgent: Final Step**

**Question:** Which film has the director died first, *Is There Justice?* or The Barrier Of Flames?
**Latest Memory:** ...The section does not provide any new information about the directors of *Is There Justice?* or The Barrier Of Flames. We would need to look for additional information to answer this question. However, we now know that *The Barrier of Flames* is a 1914 film, which is earlier than *Is There Justice?* (1931), and Chris Marker, a director who died in 2012, is mentioned. Therefore, we can infer that the director of The Barrier of Flames, Jack Harvey, died before the director of Is There Justice?, Stuart Paton.

---

**Answer:** Therefore, the answer is **The Barrier of Flames**.

---

As illustrated in the illustration boxes below, MemAgent processes the context purely sequentially. In Step 2, it correctly identifies that *Is There Justice?* was directed by Stuart Paton. However, because the immediate context chunk does not contain Paton's date of death, MemAgent updates its memory with a passive note: "The section does not provide... death dates." Crucially, as the model proceeds to Step 3 to read about the second film (*The Barrier of Flames*, directed by Jack Harvey), it suffers from forward-only processing constraint. Without a mechanism to look back or search for the missing data regarding the first director, it attempts to infer the answer from irrelevant entities present in the current chunk (e.g., confusing the target with a different director mentioned in the text, Chris Marker). Consequently, MemAgent relies on hallucinated reasoning to force a conclusion, and yields an incorrect prediction.

**The Constraint of Forward-Only Processing** The "Forward-Only" limitation prevents the agent from retrospectively attending to past information once the relevance of that information becomes clear in a later time step. This is explicitly demonstrated in the disconnect between Step 1 and Step 3.

- **The Missed Evidence (Step 1):** In Chunk 10 (Step 1), the text explicitly provides the death date of Stuart Paton (16 December 1944). However, the agent fails to recognize its relevance at this stage, as it has not yet identified Paton as the director of *Is There Justice?*.

- **The Delayed Context (Step 3):** It is not until Step 3 (Chunk 109) that the agent learns Is There Justice? was directed by Stuart Paton.

- **The Inability to Revisit:** Because the architecture processes the stream unidirectionally ($t \rightarrow t + 1$), the agent cannot revisit Chunk 10 to extract Paton's death date after recognizing his relevance in Step 3. This temporal misalignment of evidence leads to a critical information gap.

**Information Loss through Overwriting** The case further demonstrates how the finite memory bottleneck leads to the overwriting of potential knowledge with erroneous inferences. As the agent struggles to maintain a coherent narrative across long contexts, valid query states are overwritten by noisy, local correlations.

- **Accumulated Error (Step 4):** By Step 4, the agent's memory acts as a bottleneck. Since the model have already lost the grounded truth about Stuart Paton (due to the forward-only failure), the agent is forced to rely on immediate, local context.

- **The Overwriting Happens:** In Chunk 158, the agent encounters irrelevant information about Chris Marker. Desperate to resolve the query, the agent constructs a spurious logical connection, inferring that because Chris Marker died in 2012 and the films have different release dates, a comparison can be made.

- **The Result:** This hallucination overwrites the agent's previous state of "searching for information." The memory transitions from a neutral state of missing data to a confident, false state (Memory 4: "...we can infer that... Jack Harvey, died before... Stuart Paton"). This false memory effectively increases the difficulty of correctly identifying the directors' lifespans, and ultimately leads to the incorrect answer in the Final Step.

### B.5.2 CASE STUDY OF REMEMR1

---

**ReMemR1: Step 1**

**Chunk (Document 10):** ...Stuart Paton (23 July 1883 – 16 December 1944) was a British director, screenwriter and actor of the silent era.He directed 67 films between 1915 and 1938...
**Recalled Memory:** No memory was recalled.

---

**Memory 1:** ...**Stuart Paton (23 July 1883 – 16 December 1944) was a British director, screenwriter and actor of the silent era. He directed 67 films between 1915 and 1938. He also wrote for 24 films between 1914 and 1927**...
**Callback 1:** who's the director of *Is There Justice?* and *The Barrier Of Flames?*

---

**ReMemR1: Step 2**

... (nothing important in the chunk) ...
**Recalled Memory** *(who's the director of Is There Justice? and The Barrier Of Flames?)*: ...Stuart Paton (23 July 1883 – 16 December 1944) was a British director, screenwriter and actor of the silent era...

---

**Memory 2:** ...Stuart Paton (23 July 1883 – 16 December 1944) was a British director, screenwriter and actor of the silent era. He directed 67 films between 1915 and 1938. He also wrote for 24 films between 1914 and 1927...
**Callback 2:** who's the director of *Is There Justice?* and *The Barrier Of Flames*?

---

**ReMemR1: Step 3**

**Recalled Memory** *(who's the director of Is There Justice? and The Barrier Of Flames?)*: ...Stuart Paton (23 July 1883 – 16 December 1944) was a British director, screenwriter and actor of the silent era...
**Chunk (Document 83):** ...John JosephHarvey, also known as **Jack Harvey( September 16, 1881 in Cleveland, Ohio- November 9, 1954 in Los Angeles, California**) was an American film actor...
**Chunk (Document 109):** ...*Is There Justice?* **is a 1931 American crime film directed by Stuart Paton**... It is now considered a lost film...

**Memory 3:** Stuart Paton (23 July 1883 – 16 December 1944) was a British director, screenwriter and actor of the silent era. He directed 67 films between 1915 and 1938. He also wrote for 24 films between 1914 and 1927...***Is There Justice?* is a 1931 American crime film directed by Stuart Paton** and starring Rex Lease, Henry B. Walthall and Blanche Mehaffey...**Jack Harvey( September 16, 1881 in Cleveland, Ohio- November 9, 1954 in Los Angeles, California**) was an American film actor
**Callback 3:** who's the director of *The Barrier Of Flames*?

**ReMemR1: Step 4**

**Recalled Memory** *(who's the director of The Barrier Of Flames?)*: ...Stuart Paton (23 July 1883 – 16 December 1944) was a British director, screenwriter and actor of the silent era...
**Chunk (Document 154):** ...*The Barrier of Flames* **is a 1914 American short silent drama film, directed by Jack Harvey**. It stars Shep the Dog, Helen Badgley...
**Chunk (Document 158):** ...Chris Marker( 29 July 1921 – 29 July 2012) was a French writer, photographer, documentary film director...

**Memory 4:** Stuart Paton (23 July 1883 – 16 December 1944) was a British director, screenwriter and actor of the silent era. He directed 67 films between 1915 and 1938. He also wrote for 24 films between 1914 and 1927...*Is There Justice?* is a 1931 American crime film directed by Stuart Paton and starring Rex Lease, Henry B. Walthall and Blanche Mehaffey...***The Barrier of Flames* is a 1914 American short silent drama film, directed by Jack Harvey**...
**Callback 4:** who's the director of *The Barrier Of Flames*?

**ReMemR1: Step 5**

... (nothing important in the chunk) ...
**Recalled Memory** *(who's the director of The Barrier Of Flames?)*: ...*The Barrier of Flames* is a 1914 American short silent drama film, directed by Jack Harvey...

**Memory 5:** Stuart Paton (23 July 1883 – 16 December 1944) was a British director, screenwriter and actor of the silent era. He directed 67 films between 1915 and 1938. He also wrote for 24 films between 1914 and 1927...*Is There Justice?* is a 1931 American crime film directed by Stuart Paton and starring Rex Lease, Henry B. Walthall and Blanche Mehaffey...*The Barrier of Flames* is a 1914 American short silent drama film, directed by Jack Harvey...**Jack Harvey (born 1883 – died 1954) was an American director, screenwriter, and actor**...
**Callback 5:** who's the director of *The Barrier Of Flames*?

**ReMemR1: Final Step**

**Question:** Which film has the director died first, *Is There Justice?* or The Barrier Of Flames?
**Recalled Memory** *(who's the director of The Barrier Of Flames?)*: ...*The Barrier of Flames* is a 1914 American short silent drama film, directed by Jack Harvey...
**Latest Memory:** Stuart Paton (23 July 1883 – 16 December 1944) was a British director, screenwriter and actor of the silent era. He directed 67 films between 1915 and 1938. He also wrote for 24 films between 1914 and 1927...*Is There Justice?* is a 1931 American crime film directed by Stuart Paton and starring Rex Lease, Henry B. Walthall and Blanche Mehaffey...*The Barrier of Flames* is a 1914 American short silent drama film, directed by Jack Harvey...Jack Harvey (born 1883 – died 1954) was an American director, screenwriter, and actor...

**Answer:** From the memory, we can see that Stuart Paton, the director of *Is There Justice?*, died in 1944. Jack Harvey, the director of *The Barrier Of Flames*, died in 1954. Therefore, Stuart Paton died first. **Is There Justice?**

As demonstrated in the inference case, ReMemR1 successfully identifies the correct answer (*Is There Justice?*) by employing callback-enhanced memory and non-linear reasoning. These mechanisms directly address the structural failures observed in the baseline model.

**Overcoming "Forward-Only" Constraints via Callbacks**    Unlike the baseline, which discarded Stuart Paton's death date in Step 1 because its relevance was not yet known, ReMemR1 preserves this critical evidence through its active Callback Mechanism.

- **Proactive Retention (Step 1):** In Step 1, even though the connection to the film *Is There Justice?* has not yet been established, ReMemR1 records Stuart Paton's biographical data in Memory 1. Crucially, it generates a Callback query: "who's the director of *Is There Justice?*...". This signals to the model that information related to this director may become relevant later.

- **Non-Linear Bridging (Step 3):** When the agent finally encounters the film title in Step 3, it does not need to backtrack to the raw text of Step 1. Instead, the relevant information (Paton's death date) is bridged naturally from the memory history. The agent instantly links the new evidence (Film A = Paton) with the retained evidence (Paton = died 1944), effectively bypassing the limitations of forward-only processing.

**Mitigating Information Loss via Selective Retrieval**    The baseline model suffer from "memory overwriting," where early facts are overwritten by later, irrelevant noise (e.g., the Chris Marker hallucination). ReMemR1 prevents this through Selective Retrieval.

- **Robust State Maintenance (Step 4 & 5):** Instead of relying on a single, degradable memory state, ReMemR1 utilizes a retrieval mechanism. In Step 4, the Recalled Memory field explicitly retrieves the previously stored facts about Stuart Paton while simultaneously processing the new facts about Jack Harvey.

- **Noise Filtering:** By selectively recalling only the data relevant to the active Callbacks, ReMemR1 filters out the noise that confused the baseline. It ignores the irrelevant "Chris Marker" segment once the segment is passed, and focuses on the verified facts about both directors.

In the end of the inference (*e.g.,* final step), ReMemR1 successfully synthesizes evidence across distant timesteps (Step 1 and Step 3) in the reversed order, and ultimately reaches the correct inference: Stuart Paton (died 1944) died before Jack Harvey.

### B.5.3    FAILURE ANALYSIS OF REMEMR1

In this section, we conduct analysis on two error cases ReMemR1 have made on HotpotQA to reveal specific vulnerabilities in its **recall query generation** and **memory update** policies.

**Failure Pattern 1: Recall Mechanism Collapse**    In this failure mode, the agent fails to generate contextually relevant queries when faced with information gaps. Instead of formulating a targeted question to retrieve missing information, the model falls back to irrelevant queries (e.g., asking about the US President, which is the example used in the system prompt).

Consequently, the *recalled_memory* field is not populated with relevant historical context that could link "Liberal Conservative" (the ground truth style) with the specific party names found earlier. The model ultimately reaches a final answer that focuses on the name ("People's Party") rather than the ideology. This could be partly because of the recall mechanism failed to retrieve the specific semantic constraints requested by the problem.

---

**Failure Case 1: Recall Degeneration**

**Problem:** What kind of political party that combined conservative policies with liberal stances were Pio Cabanillas Gallas and Jose Maria de Areilza the leaders of?

**Step 1 Recall:** who's Pio Cabanillas Gallas and what political party did he lead? ✓(Relevent)

---

**Chunk 2:** ...Spanish Social Reform... dissolved in 1977... Manuel Cantarero would join the Liberal Citizens Action...

**Memory 2:** ...Long copy-paste of document text regarding Spanish Social Reform and United National Party...

**Callback 2:** *<recall>who's the president of the United States?</recall>* (Failure)

---

**Chunk 3 (Doc 91):** ...People's Party (Spanish: "Partido Popular"; PP) was a Spanish liberal conservative political party... The leaders of the PP were Pío Cabanillas Gallas and José María de Areilza.

**Callback 3:** *<recall>who's the president of the United States?</recall>* (Failure)

---

. . .

---

**Final Answer:** People's Party

**Ground Truth:** Liberal Conservative

---

**Failure Pattern 2: Memory Polution**    The second pattern illustrates a first impression bias, which results in a "Memory Polution Mechanism". If ReMemR1 hallucinates or misinterprets information in an early step and writes it into memory, this "poluted" memory could appear over and over again in the later steps, thus overrides correct evidence encountered. The agent fails to use the memory overwrite nor memory callback mechanism to correct previous falsehoods.

In this case, the model misinterprets a list of cars used in a race as a list of cars manufactured by Holden. This hallucination was immediately written into the new memory update. In the following step 2, the model explicitly read Document 92 which lists the correct models (sedan, station wagon). However, instead of performing a corrective update to remove the false Ford/Toyota claim, it simply appends the new text. When generating the final answer in the end, the model prioritizes the incorrect "manufactures Ford and Toyota" statement it generated earlier over the implicit correct evidence found later.

---

**Failure Case 2: Memory Polution**

**Problem:** This Australian racing driver currently drives the No. 22 Holden VF Commodore produced by Holden Commodore which manufactures what other models? (Ground Truth: *sedan and station wagon*)

**Chunk 1:** ...The Virgin Australia Supercars Championship is a motorsport series in Australia that uses Holden VF Commodore, Ford Falcon, and Toyota Supra models...

**Memory 1:** ...The Holden VF Commodore is a model produced by Holden Commodore, **which also manufactures the Ford Falcon and Toyota Supra models.**

---

**Chunk 3:** ...The Holden Commodore (VF)... Its range includes the sedan and station wagon variants...

**Memory 3:** ...The Holden VF Commodore is a model produced by Holden Commodore, which also manufactures the Ford Falcon and Toyota Supra models... Its range includes the sedan and station wagon variants that sell under the luxury Holden Calais (VF) nameplate

**Callback 3:** *who's the president of the United States?* (Irrelevant)

---

. . .

---

**Final Answer:** Ford Falcon, Toyota Supra

**Ground Truth:** Sedan and Station Wagon

---

## C IMPLEMENTATION DETAILS

### C.1 FULL EXPRESSION OF TRAINING OBJECTIVE

Our model is optimized with a variant of GRPO objective. The full expression of our training objective can be written as:

$$
\arg\max_{\theta} J_{\text{GRPO}}(\theta) = \mathbb{E}_{(Q,Y),\{\tau^{(g)}\}_{g=1}^{G} \sim \pi_{\theta_{\text{old}}}} \left[ \frac{1}{G(T+1)} \sum_{g=1}^{G} \sum_{t=1}^{T+1} \frac{1}{|s_t^{(g)}|} \sum_{i=1}^{|s_t^{(g)}|} \min \left( \rho_{t,i}^{(g)} \hat{A}_t^{(g)}, \right. \right.
$$
$$
\left. \left. \text{clip}\left( \rho_{t,i}^{(g)}, 1-\epsilon, 1+\epsilon \right) \hat{A}_t^{(g)} \right) - \beta \, \mathbb{D}_{\text{KL}}[\pi_\theta \,\|\, \pi_{\text{ref}}] \right],
$$

(10)

where $\rho_{t,i}^{(g)}$ is the importance sampling ratio:

$$
\rho_{t,i}^{(g)} = \frac{\pi_\theta \left( s_{t,i}^{(g)} \mid s_{t,<i}^{(g)}, s_{<t}^{(g)}, Q, c_{t-1} \right)}{\pi_{\theta_{\text{old}}} \left( s_{t,i}^{(g)} \mid s_{t,<i}^{(g)}, s_{<t}^{(g)}, Q, c_{t-1} \right)}.
$$

(11)

Here, $s_{t,i}^{(g)}$ denotes the $i$-th token in the $t$-th state of trajectory $g$, $\epsilon$ is the clipping ratio, $\beta$ is the KL coefficient, and $\hat{A}_t^{(g)}$ is the normalized advantage. We assume $c_T = \varnothing$ for notational convenience.

### C.2 TRAINING HYPERPARAMETERS

The training of ReMemR1 was built upon the verl[1] framework, with efficient trajectory generation powered by the sglang[2] engine. We employed Fully Sharded Data Parallelism (FSDP) for distributed training, and used `bfloat16` precision for both training and evaluation. Table 8 summarizes the primary hyperparameters used in our method.

Although we evaluated the model with varying numbers of context documents during testing, the training setup consistently used 200 documents per sample, resulting in approximately 30K input tokens. Each document chunk $c_t$ was limited to a maximum length of 5000 tokens, yielding $T \approx 6$ during training. At each timestep and at the final state, the model generated rollouts with a temperature of 1, up to a maximum of 2048 tokens.

The 3B version of ReMemR1 and its variants are trained on 16 H800 GPUs and converge after 100 hours. The 7B model is trained on 32 H800 GPUs, reaching convergence after 80 hours.

Table 8: Primary hyperparameters used in training.

| Hyper-parameter | Value |
|---|---|
| Training Batch Size | 128 |
| Micro Training Batch Size | 8 |
| Total Converge Steps | 200~300 |
| Actor Model Learning Rate | $1 \times 10^{-6}$ |
| Actor Model Warmup Steps | 20 |
| Rollout Temperature | 1 |
| Max Chunk Length | 5000 |
| Training Chunk Number T | 6 |
| Max Response Length | 2048 |
| KL Coefficient $\beta$ | 0.001 |
| Clip Ratio $\epsilon$ | 0.2 |
| Group Size $G$ | 16 |

---

[1] https://github.com/volcengine/verl
[2] https://github.com/sgl-project/sglang

## C.3 EVALUATION SETTINGS

To ensure the challenging nature of the samples, we only use samples from the hard difficulty level for training. Questions in these datasets typically require at least two pieces of evidence to answer, and there exist dependencies between the evidence. Due to the extraordinary computational cost of long-context QA, we subsample 128 samples from each benchmarks with a random seed of 4, following Yu et al. (2025a).

## D COMPLEXITY ANALYSIS

In this section, we analyze the computational complexity of ReMemR1 and show that it preserves the linear complexity of conventional memory-agent approaches.

### D.1 BASELINE COMPLEXITY

In the "memorize while reading" paradigm, the agent processes a sequence of $T$ document chunks $\{c_1, c_2, \ldots, c_T\}$ in order. At each step $t$, it updates the memory via:

$$m_{t+1} = \pi(Q, c_t, m_t). \tag{12}$$

Each update requires $O(1)$ memory operations and a constant number of forward passes through the policy network. Thus, the overall time complexity is $O(T)$. The space requirement is a summation of the document chunks and the memory at each step, which is $O(T + 1) = O(T)$ in total.

### D.2 COMPLEXITY OF REMEMR1

ReMemR1 augments the state by including a query component $q_t$ and a retrieval function $\mathcal{E}$ over past memories:

$$s_{t+1} = (m_{t+1}, q_{t+1}) = \pi(Q, c_t, m_t, \mathcal{E}(\{m_i\}_{i \leqslant t}, q_t)). \tag{13}$$

This paradigm also performs the same number of state transition, which is $O(n)$ times of LLM generation. Compared to Eq. 12, our method includes two sources of computational overhead:

- **Storage of previous memories.** Although the state transition references $\{m_i\}_{i \leqslant t}$, each $m_i$ is itself a fixed-length vector (e.g., the hidden state of the model). Maintaining this list across $T$ steps requires $O(T)$ additional space. This is the same order as storing the original text chunks, but with a smaller constant term.

- **Retrieval operation** The retrieval function $\mathcal{E}$ computes similarity between $q_t$ and past memory states. If implemented with exact maximum similarity search over $\{m_i\}_{i \leqslant t}$, the cost per step could be $O(t)$. However, in practice, we use lightweight recall-based heuristics or an index that supports sublinear approximate nearest neighbor search. This operation is negligible compared against the consumption of the state transition model $\pi_\theta$, which is often a 3B or 7B level LLM. Thus, the total cost across $T$ steps remains $O(T)$ in expectation.

Therefore, ReMemR1 preserves the same asymptotic $O(T)$ time and $O(T)$ space complexity as the conventional memory-agent paradigm, while substantially enhancing the agent's ability to perform non-linear reasoning through retrieval.

## E PROMPT TEMPLATE

We use separate prompt templates for the generation of intermediate states $s_{1 \leqslant t \leqslant T}$ and the final states $s_{T+1}$. The prompts are listed below:

---

**Prompt Template for Intermediate States.**

You are presented with a problem, a section of an article that may contain the answer to the problem, and a previous memory. You should generate a response in the following format:
- Output your thinking process in `<thinking>your_thinking_process</thinking>`. - Read the provided section carefully and update the memory with the new information that helps to answer the problem in only one `<update>the_updated_memory</update>` action. Be sure to retain all relevant details from the previous memory while adding any new, useful information.
- If you notice partial key evidence that is not enough to answer the problem, also output only one `<recall>query</recall>` (e.g. "`<recall>who's the president of the United States?</recall>`") to retrieve information in previous memories.

<problem> QUESTION </problem>
<recalled_memory> RECALLED MEMORY </recalled_memory>
<memory> MEMORY </memory>
<section> DOCUMENT CHUNK </section>

Updated memory:

---

**Prompt Template for Final States.**

You are presented with a problem and a previous memory. Please answer the problem based on the previous memory and put the answer in \boxed{}.

<problem> QUESTION </problem>
<recalled_memory> RECALLED MEMORY </recalled_memory>
<memory> MEMORY </memory>

Your answer:

---

## F   THE USE OF LARGE LANGUAGE MODELS

In the preparation of this manuscript, we utilized an LLM as a writing assistance. The use of the LLM was limited to proofreading for grammatical errors, checking for typos, and improving the clarity and readability of existing text. The LLM was not used for any core intellectual contributions, including but not limited to research ideation, formulation of the methodology, analysis of results, or drafting of the original manuscript. All scientific claims, arguments, and the final text are the sole work of the human authors, who pay full responsibility for all content.

