# OpenReview forum: "Look Back to Reason Forward: Revisitable Memory for Long-Context LLM Agents"
_ICLR.cc/2026/Conference — ICLR 2026 Poster_

### Official Review · Reviewer_VkNG · 2025-10-27

**Soundness:** 3
**Presentation:** 3
**Contribution:** 3
**Rating:** 4
**Confidence:** 4

**Summary:**

This paper presents ReMemR1, a memory-augmented large language model (LLM) agent for long-context question answering (QA). The key innovation is a callback-enhanced memory mechanism that enables the model to revisit and retrieve earlier memory states instead of relying solely on a forward-only “memorize while reading” paradigm. To address the sparse reward issue commonly faced in reinforcement learning (RL)-based memory agents, the authors propose Reinforcement Learning with Multi-Level Rewards (RLMLR), which combines trajectory-level (final-answer) and step-level (intermediate) rewards. Experiments on two datasets demonstrate consistent improvements over strong baselines such as MemAgent and Qwen2.5, particularly under long-context and distant-evidence settings.

**Strengths:**

- The paper is clearly written, well-organized, and easy to follow, which facilitates understanding of the technical contributions.

- The proposed method achieves consistent gains on both in-distribution and out-of-distribution datasets, with notable advantages in challenging long-context scenarios.

- The idea of non-linear reasoning via callback-based memory is conceptually interesting and potentially useful for improving long-context reasoning.

**Weaknesses:**

- While the callback mechanism is intuitively appealing, it can be viewed as an incremental extension of existing memory-retrieval or recurrent-state models. The distinction from prior works (e.g., MemAgent) is not sufficiently emphasized.

- The paper provides limited formal or analytical insight into *why* the proposed retrieval and multi-level reward structure lead to improved reasoning or generalization.

- The authors argue that existing memory-based agents suffer from irreversible forward-only processing and information loss because they cannot revisit previous states. However, this argument is not entirely convincing. When the policy updates its memory at each step, it already considers both the previous memory and the current input (query or document chunk), which inherently allows recursive integration of prior information. It is unclear why an additional “memory recall” mechanism is strictly necessary, rather than improving the existing update function to retain or reweight key information.

- The proposed design retrieves relevant content from all previous memory states, which implies that the model must maintain access to the entire memory history. This could be computationally expensive and may not scale to extremely long contexts. The paper should clarify whether ReMemR1 explicitly stores all intermediate memories or uses a compressed or differentiable representation to manage memory growth.

- Important implementation aspects, such as retrieval efficiency, computational overhead, and inference latency, are under-discussed. It remains unclear how the proposed approach scales in both training and inference compared to other memory-based methods.

- Additional baselines, especially recent long-context or memory-augmented LLMs, would make the evaluation more comprehensive and convincing.

**Questions:**

1. How does ReMemR1 perform when the memory corpus exceeds the retrieval limit or available storage capacity? Or how does other methods enlarge their memory storage?

2. Does the system employ any pruning or compression strategy to prevent unbounded memory growth?

---

> ### Author Response · Authors · 2025-11-21
> **Author Response to Reviewer VkNG (1/4)**
>
> We thank the reviewer for the time and effort in reviewing this paper.
>
> > **Weakness 1.** While the callback mechanism is intuitively appealing, it can be viewed as an incremental extension of existing memory-retrieval or recurrent-state models. The distinction from prior works (e.g., MemAgent) is not sufficiently emphasized.
>
> Thank you for highlighting this point. We appreciate your recognition over proposed callback mechanism, and we also agree with your concern that its distinction from prior memory-retrieval or recurrent-state models was not clearly articulated.
>
> **One-Sentence Summary:** We attempt to introduce the idea of **memory retrieval** into **recurrent-state models** to mitigate irreversible processing and progressive information loss issues.
>
> To clarify the confusion caused by our insufficient explanation in the paper, we prepared the following table comparing the notions of memory retrieval models, recurrent-state models, and our method.
> Here, $L_{\text{int}}$ denotes the length of internal memory (generated during inference), $L_{\text{ext}}$ denotes the length of external memory (from knowledge bases or memory stores), and $N$ is the number of recurrent steps in recurrent-state models.
>
> **Table 1.** Comparison between memory retrieval models, recurrent-state models, and our method.
>
> | | Memory Retrieval Models | Recurrent-State Models | Our Method |
> | -- | -- | -- | -- |
> | Overview | (1) Store all documents/dialogues as memory (2) Retrieve memories on demand during inference | (1) Process context chunk-by-chunk (2) Maintain only the latest memory state | (1) Process context chunk-by-chunk (2) Maintain *all* memory states (3) Retrieve memories on demand during inference |
> | Use Cases | Cross-Dialogue Memory in Chatbots, RAG | Long-Context QA | Long-Context QA |
> | Representative Works | MemOS, Search-R1 | MemAgent, MemAlpha | ReMemR1 |
> | Benefits | No progressive information loss; low GPU computation | Low computational cost | Addresses irreversible processing and progressive information loss |
> | Drawbacks | High memory usage since $L_{\text{ext}}$ can be large | Progressive information loss <br> Irreversible processing | Quadratic computation scaling with respect to $N$ |
> | CPU Computation | $\mathcal{O}(L_{\text{int}} + L_{\text{ext}})$ | $\mathcal{O}\!\left(N \cdot L_{\text{int}} + L_{\text{ext}}\right)$ | $\mathcal{O}\!\left(N^{2} \cdot L_{\text{int}} + L_{\text{ext}}\right)$ |
> | Memory Usage per Sample | $\mathcal{O}(L_{\text{ext}} + L_{\text{int}})$ | $\mathcal{O}(L_{\text{int}} + L_{\text{ext}} / N)$ | $\mathcal{O}(N \cdot L_{\text{int}} + L_{\text{ext}} / N)$ |
> | GPU Computation | $\mathcal{O}(L_{\text{int}}^3)$ | $\mathcal{O}\!\left(N \cdot \left(\frac{L_{\text{ext}}}{N} + L_{\text{int}}\right)^2 \, L_{\text{int}}\right)$ | $\mathcal{O}\!\left(N \cdot \left(\frac{L_{\text{ext}}}{N} + L_{\text{int}}\right)^2 \, L_{\text{int}}\right)$ |
>
>
> Importantly, in typical recurrent-state models, $L_{\text{int}}$ is usually one to two orders of magnitude smaller than $L_{\text{ext}}$. Motivated by this, **our method trades some theoretical complexity for improved accuracy**, while keeping the actual overhead minimal thanks to the small constant factor $L_\text{int}$. Our experiments indicate that the memory callback mechanism introduces only marginal additional time and space overhead in practice.
>
> We are now integrating these experiments and analyses into the revised manuscript and will soon update the manuscript PDF.

---

> ### Author Response · Authors · 2025-11-21
> **Author Response to Reviewer VkNG (2/4)**
>
> > **Weakness 2.** The paper provides limited formal or analytical insight into why the proposed retrieval and multi-level reward structure lead to improved reasoning or generalization.
>
> Thank you for pointing out the lack of formal and analytical discussion on how our method behaves. Following your suggestion, we now provide additional analysis for both the memory callback and the multi-level reward components.
>
> 1. Why does the proposed **memory callback** lead to improved reasoning?
>
> We conduct **additional experiments** to specifically demonstrate how the memory callback helps mitigate the *irreversible processing* issue in prior recurrent-state methods.
>
> To analyze this effect, we compare **MemAgent-7B** and **ReMemR1-7B** under two settings:
> - **Vanilla document order**: documents are randomly shuffled, as in Table 1 of the paper.
> - **Reversed order**: multi-hop evidence is given in reverse order, e.g., 3rd-hop evidence first, then 2nd-hop, then 1st-hop.
>
> The results are reported in **Table 2** below. We observe that performance drops caused by reversing the evidence order are substantial for MemAgent (typically around 2–3 points, up to 3.5), but are significantly alleviated for ReMemR1 (typically around 0–1.7 points; some negative “drops” indicate slight gains under the reversed setting).
> From these findings, we obtain the analytical insight that **ReMemR1 improves long-context reasoning robustness by alleviating the irreversible processing phenomenon**, thanks to its ability to revisit and reuse earlier memory states via memory callback.
>
> **Table 2.** Long-context QA results on the HotpotQA benchmark with (1) random evidence orders and (2) reversed evidence orders.
>
> | | Number of Docs | 50 | 100 | 200 | 400 | 800 | 1600 | 3200 | 6400 |
> | -- | -- | -- | -- | -- | -- | -- | -- | -- | -- |
> | ReMemR1-7B | Vanilla | 82.30 | 82.80 | 81.10 | 78.90 | 82.00 | 79.70 | 80.00 | 80.80 |
> | | Reversed Evidence | 81.62 | 82.96 | 79.36 | 80.41 | 81.88 | 78.23 | 81.01 | 78.30 |
> | | Performance Drop | 0.68 | -0.16 | 1.74 | -1.51 | 0.12 | 1.47 | -1.01 | 2.50 |
> | MemAgent-7B | Vanilla | 81.80 | 78.90 | 78.90 | 77.00 | 79.70 | 72.10 | 74.00 | 75.80 |
> | | Reversed Evidence | 79.24 | 76.78 | 75.80 | 74.88 | 76.82 | 69.80 | 71.16 | 72.30 |
> | | Performance Drop | 2.56 | 2.12 | 3.10 | 2.12 | 2.88 | 2.30 | 2.84 | 3.50 |
>
>
> 2. Why does the **multi-level reward** yield improvements?
>
> In multi-turn LLM agents, the agent executes an action at *every step*, but typically only receives a reward signal at *the final step*. Naively propagating this final-step reward to all previous steps leads to **sparse and noisy credit assignment**: it becomes difficult to accurately assess which intermediate actions were truly helpful [1, 2].
>
> In the recurrent-state setting considered in our work, we make a key observation about the structure of the intermediate states. For different trajectories $g$, the intermediate state can be written as $s_t^{(g)} = (Q, c_t, m_t^{(g)})$ , where $Q$ (the question) and $c_t$ (the current context) are **shared across trajectories at the same step $t$**, which makes we can isolate the contribution of the memory $m_t^{(g)}$ by measuring the information gain brought by the model-generated memory. As a result, the multi-level reward can **accurately encourage behaviors that “memorize better”**. Table 2 in the paper empirically supports this insight: it shows that choosing an appropriate value of the multi-level reward coefficient $\alpha$ consistently improves performance.
>
> References:
>
> [1] Reinforcing Multi-Turn Reasoning in LLM Agents via Turn-Level Reward Design
>
> [2] Multi‑level Advantage Credit Assignment for Cooperative Multi‑Agent Reinforcement Learning

---

> ### Author Response · Authors · 2025-11-21
> **Author Response to Reviewer VkNG (3/4)**
>
> > **Weakness 3.** The authors argue that existing memory-based agents suffer from irreversible forward-only processing and information loss because they cannot revisit previous states ... It is unclear why an additional “memory recall” mechanism is strictly necessary, rather than improving the existing update function to retain or reweight key information.
>
> Thanks for pointing out our weak spot at demonstrating the necessity of the additional “memory recall” mechanism.
>
> At your suggestion, we prepare **new case studies** that help demonstrate why the vanilla recurrent-state paradigm could fail under certain situation, and how does the model exactly benefits from the additional “memory recall” mechanism.
> We are working on a "B.3 Case Study" subsection in the appendix.
>
>
> > **Weakness 4.** ... The paper should clarify whether ReMemR1 explicitly stores all intermediate memories or uses a compressed or differentiable representation to manage memory growth.
> >
> > **Weakness 5.** ... It remains unclear how the proposed approach scales in both training and inference compared to other memory-based methods.
>
> We thank the reviewer for pointing out our problem. **Yes, ReMemR1 explicitly stores all intermediate memories.**
>
> We also fully concur with your concern - ReMemR1 does not implement a strategy to prevent the memory growth, yet a quantitative discussion over the computational overhead is necessary.
> We have conducted **new control experiment** to measure the time and memory complexity of our method.
>
> We report the following metrics on the HotpotQA benchmark at inference: (1) Accuracy (2) Time Complexity: Time/Sample and Callback TIme/Sample (2) Space Complexity: Total Memory and Callback Memory. The results are shown in Table 3.
>
> **Table 3.** Comparison of computational performance during **inference**.
>
> | | |Number of Docs| 50 | 100 | 200 | 400 | 800 | 1600 | 3200 | 6400 |
> | -- | -- | -- | -- | -- | -- | -- | -- | -- | -- | -- |
> | Accuracy | MemAgent | | 81.8 | 78.9 | 78.9 | 77 | 79.7 | 72.1 | 74 | 75.8 |
> | | ReMemR1 | | 82.3 | 82.8 | 81.1 | 78.9 | 82 | 79.7 | 80 | 80.8 |
> |Time|MemAgent|Time / Sample (s)|14.51|22.41|38.16|69.89|152.62|356.82|676.80|1422.17|
> ||ReMemR1|Time / Sample (s)|16.70|26.02|46.90|90.53|211.33|527.85|1004.29|1935.84|
> ||ReMemR1|Callback Time / Sample (s)|0.01|0.03|0.06|0.12|0.27|0.40|0.93|1.73|
> ||ReMemR1|Callback / Total|0.07%|0.10%|0.14%|0.14%|0.13%|0.08%|0.09%|0.09%|
> | Space | MemAgent | Total Memory (MB) | 811.95 | 818.32 | 833.18 | 862.68 | 924.18 | 1050.88 | 1358.79 | 1989.44 |
> | | ReMemR1 | Total Memory (MB) | 808.90 | 824.80 | 868.80 | 893.44 | 964.17 | 1117.41 | 1418.86 | 2005.30 |
> | | ReMemR1 | Callback Memory (MB) | 0.01 | 0.02 | 0.03 | 0.05 | 0.10 | 0.16 | 0.23 | 0.34 |
> | | ReMemR1 | Callback / Total | <0.001% | <0.001% | <0.001% | <0.001% | <0.001% | <0.001% | <0.001% | <0.001% |
>
> We also measures the (1) Time/Sample and (2) Total Memory usage during training, and the numbers are reported in Table 4.
>
> During training, we use the same context length (200 documents) for all steps, so we measures the scalability by testing the time consumption and Peak memory usage at different time steps. The results shows that the training time usage of ReMemR1 is stable even after 100 steps of training; the memory usage overhead is also acceptable.
>
> **Table 4.** Comparison of computational performance during **training**.
>
> | | Average Time (s) | Step 1 (s) | Step 10 (s) | Step 100 (s) | Peak Memory Usage (GB) |
> | -- | -- | -- | -- | -- | -- |
> | MemAgent | 1247.1715 | 1278.8475 | 1366.9076 | 1377.2914 | 124.97 |
> | ReMemR1 | 1467.7195 | 1463.2507 | 1518.9967 | 1456.6905 | 131.15 |
>
> From the results, we see that: (1) the retrieval steps yield little time and space overhead compared to the transformer forwarding (2) ReMemR1 indeed costs more time compared to baseline method, which can be attributed to the generation of new "< recall >callback query< /recall >" tokens.
> It can also be observed that ReMemR1 successfully "trades improved accuracy with time/space complexity": it **yields up to 5% higher accuracy with acceptable overhead; among this, the cost of memory callback only accounts for a very small part (<0.2% time and <0.001% space).**

---

> ### Author Response · Authors · 2025-11-21
> **Author Response to Reviewer VkNG (4/4)**
>
> > **Weakness 6.** Additional baselines, especially recent long-context or memory-augmented LLMs, would make the evaluation more comprehensive and convincing.
>
> We thank the reviewer for the suggestion of introducing more baseline methods, especially recent long-context or memory-augmented LLMs.
> We've conduct experiments with **additional memory-augmented baselines**:
> - MemOS, with GPT-4 as the backbone LLM
> - MemAlpha (official 4B checkpoint). **the experiments for this baseline are still running, and the results will be updated once we have them.**
> - Search-R1 (official 7B-PPO checkpoint)
> And the results are reported in Table 5:
>
> **Table 5.** Performance comparison with additional baselines.
>
> ||Number of Docs|50|100|200|400|800|1600|3200|6400|
> |---|---|---|---|---|---|---|---|---|---|
> |HotpotQA|R1-Distill|40.6|25.8|10.2|0.8|1.6|2.3|1.5|3.1|
> ||Qwen2.5-1M|75.8|71.9|68.0|67.2|69.5|54.7|22.7|0.0|
> ||MemAgent|81.8|78.9|78.9|77.0|79.7|72.1|74.0|75.8|
> ||MemOS w/ GPT-4|70.0|62.0|58.8|57.6|68.1|64.8|60.6|63.7|
> ||Search-R1|32.4|38.0|39.3|38.5|41.6|40.3|41.2|40.5|
> ||MemAlpha|||||||||
> ||Ours|82.3|82.8|81.1|78.9|82.0|79.7|80.0|80.8|
> |2WikiMultiHotQA|R1-Distill|36.7|29.7|25.8|0.0|0.8|2.3|2.3|0.8|
> ||Qwen2.5-1M|62.5|59.4|57.8|47.7|46.1|45.3|25.8|0.0|
> ||MemAgent|61.7|57.8|50.8|47.6|50.7|44.5|46.9|44.7|
> ||MemOS|41.5|37.1|33.1|26.8|29.8|29.1|28.6|27.9|
> ||Search-R1|32.6|37.8|39.6|37.0|41.8|40.4|40.7|41.0|
> ||MemAlpha|||||||||
> ||Ours|63.9|63.1|55.6|54.5|54.7|45.4|48.9|50.3|
>
> We observe that:
> - MemOS with GPT-4 yields results slightly worse than memorize-while-reading baseline and our method.
> - Search-R1 also performs sub-optimal, while the results align with the ones in their paper. The performance gap are caused by only the retrieved documents are provided to the LLM, but the full context are not exactly visible.
>
> > **Quesion 1.** How does ReMemR1 perform when the memory corpus exceeds the retrieval limit or available storage capacity? Or how does other methods enlarge their memory storage?
> >
> > **Question 2.** Does the system employ any pruning or compression strategy to prevent unbounded memory growth?
>
> Thank you for raising these important questions regarding memory scalability.
>
> At the current stage, **ReMemR1 does not employ any memory pruning or compression strategy**. The primary reason is that, throughout all our experiments, **we did not observe the memory footprint approaching or exceeding the available storage capacity**. Even with the full history of memory states retained, the overall memory consumption remained within a manageable range.
>
> More importantly, our experiments show that **maintaining the LLM-generated memories bring clear accuracy improvements**. Because the system benefits from keeping richer historical information, we chose not to apply compression mechanisms that might risk information loss.

---

> > ### Comment · Reviewer_VkNG · 2025-11-28
> >
> > Thank you for the further clarifications and detailed responses. They resolved several of my earlier concerns, and I will raise my previous score accordingly.

---

> > > ### Author Response · Authors · 2025-11-28
> > > **Thanks for your recognization & New Updates to the paper**
> > >
> > > Thank you very much for your constructive engagement during the rebuttal phase and for your willingness to improve the score. We appreciate your recognition of the value of our work.
> > >
> > > We have just uploaded a **revised manuscript** incorporating the changes we discussed. Specifically, we have:
> > >
> > > 1. We have **modified the introduction and the abstract** to better articulate the limitation in forward-only models, and the motivation & contribution of this work. This clarifies why a callback mechanism is strictly necessary compared to simply improving the memory update policy.
> > > 2. We have added the **computational overhead analysis** (New RQ3 in Section 3.4, table 5) to the paper. As discussed, this highlights that ReMemR1 achieves significant accuracy gains with negligible overhead.
> > > 3. We have added **six pages of new case studies and failure analysis** in our appendix B.5 and B.6, which may help better demonstrate the benefits of our method and the limitations remained.
> > >
> > > We hope these revisions further strengthen the paper, and will still work the paper (new baseline and framework rework) till the end of the rebuttal period. Thank you again for championing our work!
> > >
> > > Best regards,
> > >
> > > The Authors

---

### Official Review · Reviewer_EGXY · 2025-10-30

**Soundness:** 3
**Presentation:** 2
**Contribution:** 3
**Rating:** 6
**Confidence:** 4

**Summary:**

The paper presents a novel approach for long-text QA comprised of two components. The first component is a memory-augmented agent that stores multiple historical memories while processing the long text rather than rewriting a single memory state. It allows the agent to revisit past memories via query-based retrieval, enabling non-linear reasoning paths.  The second component is a modification of GRPO with a detailed reward function that combines traditional outcome rewards and step-level state rewards to improve the relevance of the generated reasoning sequences. The proposed method is evaluated on two MHQA datasets and shows improvement over vanilla model and a MemAgent baseline.

**Strengths:**

1. The paper provides a clear motivation for addressing the limitations of "memorize while reading" paradigm.
2. The experimental results demonstrate consistent improvements across multiple model sizes and varying context lengths.
3. The proposed RLMLR approach is clearly described.

**Weaknesses:**

1. Poor figure presentation. Throughout the paper, figures are consistently misaligned with their textual references, which significantly disrupts the reading flow and makes it difficult to follow the arguments.
2. The memory efficiency of the proposed method compared to the "memorize while reading" baseline is not addressed. The proposed approach requires storing multiple historical memory states, and it is not discussed how such large memory storage affects inference time and cost.
3. The statement about per-step reward innovation should be clarified. Per-step rewards are a common practice in RL.
4. The paper does not provide the callback mechanism failures analysis.

**Questions:**

1. How does the memory overhead scale with task complexity and document length compared to the "memorize while reading" paradigm?
2. What are the typical failure modes of the proposed retrieval mechanism?

---

> ### Author Response · Authors · 2025-11-21
> **Author Response to Reviewer EGXY (1/2)**
>
> We thank the reviewer for the time and effort in reviewing this paper.
>
> > **Weakness 1. Poor figure presentation.**  Throughout the paper, figures are consistently misaligned with their textual references, which significantly disrupts the reading flow and makes it difficult to follow the arguments.
>
> We sincerely thank the reviewer for pointing out the issues in our figure presentation. We acknowledge that the current visualizations are not ideal—especially *Figure 1*, which does not clearly convey the motivation or methodological flow of the paper. We are now improving our figure presentation by:
> 1. **completely reworking** the framework illustration
> 2. **adding** a new teaser figure to demonstrate our motivation
> 3. **refining** all other figures in this work
>
> These updates will be incorporated into the revised version within a few days.
>
> > **Weakness 2.** The memory efficiency of the proposed method compared to the "memorize while reading" baseline is not addressed. The proposed approach requires storing multiple historical memory states, and it is not discussed how such large memory storage affects inference time and cost.
> > **Question 1.** How does the memory overhead scale with task complexity and document length compared to the "memorize while reading" paradigm?
>
> We appreciate the reviewer’s emphasis on quantitatively evaluating memory overhead. Following your suggestion, we performed **both theoretical and empirical analyses** to more thoroughly compare the memory cost of our method against the "memorize while reading" paradigm.
>
> 1. **Theoretical Analysis of Memory Complexity**
>
> The theoretical comparison of complexity between our method and prior work is shown in Table 1. Here,
> - $L_{\text{ext}}$ denotes the length of external memory (i.e., context documents, 50, 100, 200, … in our experiments),
> - $L_{\text{int}}$ denotes the length of internal memory (model-generated memory), and
> - $N$ is the number of interaction steps.
>
> In our method, the key design choice is to *only store the generated memories* $L_{\text{int}}$, whose scale is **orders of magnitude smaller** than the full external context $L_{\text{ext}}$.
>
> **Table 1.** Theoretical comparison of memory usage
>
> | | "Memorize while Reading" paradigm | Our Method |
> | --- | --- | --- |
> | Overview | (1) Process context chunk-by-chunk (2) Maintain only the latest memory state | (1) Process context chunk-by-chunk (2) Maintain *all* memory states (3) Retrieve memories on demand during inference |
> | Memory Usage | $\mathcal{O}(L_{\text{int}} + L_{\text{ext}} / N)$ | $\mathcal{O}(N \cdot L_{\text{int}} + L_{\text{ext}} / N)$ |
>
> 2. **Empirical Analysis**
>
> Table 2 presents the empirical comparison of accuracy and memory usage across different context lengths.
>
> **Table 2.** Empirical comparison of accuracies and memory usage
>
> | | | Number of Docs | 50 | 100 | 200 | 400 | 800 | 1600 | 3200 | 6400 |
> | -- | -- | -- | -- | -- | -- | -- | -- | -- | -- | -- |
> | Accuracy | MemAgent | | 81.8 | 78.9 | 78.9 | 77 | 79.7 | 72.1 | 74 | 75.8 |
> | | ReMemR1 | | 82.3 | 82.8 | 81.1 | 78.9 | 82 | 79.7 | 80 | 80.8 |
> | Memory Usage | MemAgent | Total Memory (MB) | 811.95 | 818.32 | 833.18 | 862.68 | 924.18 | 1050.88 | 1358.79 | 1989.44 |
> | | ReMemR1 | Total Memory (MB) | 808.90 | 824.80 | 868.80 | 893.44 | 964.17 | 1117.41 | 1418.86 | 2005.30 |
> | | ReMemR1 | Callback Memory (MB) | 0.01 | 0.02 | 0.03 | 0.05 | 0.10 | 0.16 | 0.23 | 0.34 |
> | | ReMemR1 | Callback / Total | <0.001% | <0.001% | <0.001% | <0.001% | <0.001% | <0.001% | <0.001% | <0.001% |
>
> We find that:
> 1. Storing generated memories costs less than 1 MB per sample, which is negligible compared with the overall inference-time memory footprint.
> 2. Our method consistently outperforms the baseline without callbacks, achieving up to 5% absolute accuracy gain at 6400 documents.
>
> Thus, we conclude that our method achieves up to **20% error-rate reduction** at the cost of **<1 MB additional memory**, which is a very favorable tradeoff.

---

> ### Author Response · Authors · 2025-11-21
> **Author Response to Reviewer EGXY (2/2)**
>
> > **Weakness 3.** The statement about per-step reward innovation should be clarified. Per-step rewards are a common practice in RL.
>
> We appreciate the reviewer’s observation. Indeed, **per-step rewards are not unique to our work**. They are widely used and studied to mitigate the sparse and noisy credit assignment problem in multi-step RL, as demonstrated in prior work [1, 2].
>
> Our contribution lies not in proposing per-step rewards as a general RL concept, but in **identifying a unique structural property of the memorize-while-reading setting** that enables a more targeted and informative step-level reward design:
>
> - In our setting, different trajectories share structurally similar states at the same step $t$.
>   Specifically, for state $(m_t, c_t, Q)$, both $c_t$ and $Q$ remain identical across trajectories at step $t$.
> - This structural invariance allows us to pinpoint which actions and memory updates are truly responsible for improving downstream reasoning by only measuring $m_t$ and $q_t$.
>
> This insight guided the design of our multi-level reward scheme. We acknowledge that calling it “RLMLR” may unintentionally suggest a new RL paradigm; we will update the terminology in the revision to better reflect that it is a **novel reward design tailored for recurrent-memory reasoning**, rather than a general RL framework or algorithm.
>
> References:
>
> [1] Reinforcing Multi-Turn Reasoning in LLM Agents via Turn-Level Reward Design
>
> [2] Multi‑level Advantage Credit Assignment for Cooperative Multi‑Agent Reinforcement Learning
>
> > **Weakness 4.** The paper does not provide the callback mechanism failures analysis.
> >
> > **Question 2.** What are the typical failure modes of the proposed retrieval mechanism?
>
> We thank the reviewer for raising this important point. **We are currently working on a dedicated failure analysis section**, and it will be added to the manuscript in a few days.

---

> ### Author Response · Authors · 2025-11-28
> **New Updates to the paper & Looking forward to discuss with you**
>
> As the discussion period progresses, we want to inform you that we have uploaded a **revised manuscript** containing significant updates and new experimental data. We believe these additions strengthen the paper and may address potential questions regarding efficiency and model behavior.
>
> Summary of Key Updates:
> 1. **Clarified Motivation (Intro & Abstract):** We modified the Introduction and Abstract to better articulate the limitations of forward-only models. We specifically clarify why a callback mechanism is strictly necessary compared to simply improving the memory update policy.
>
> 2. **Computational Overhead Analysis (New RQ3, Section 3.4, Table 5):** We added a detailed efficiency analysis. The results demonstrate that ReMemR1 achieves **significant accuracy gains with negligible overhead**, addressing potential scalability concerns.
>
> 3. **Extended Qualitative Analysis (Appendix B.5 & B.6):** We added **six pages of new case studies and failure analysis**. These visualization examples demonstrate how our method successfully retrieves past evidence where baselines fail, while also transparently discussing remaining limitations.
>
> We are continuing to work on the paper (including new baselines and framework refinements) throughout the rebuttal period. We would value your thoughts on these updates and remain available to answer any questions you may have.
>
> Best regards,
>
> The Authors

---

### Official Review · Reviewer_K85P · 2025-11-03

**Soundness:** 3
**Presentation:** 2
**Contribution:** 3
**Rating:** 4
**Confidence:** 4

**Summary:**

This paper introduces ReMemR1, an LLM training paradigm designed to overcome memory-related challenges in long-context QA. Existing long-context methods process documents in a single forward pass, while memory agents may result in irreversible information loss due to the linear processing of memories. In contrast, ReMemR1 addresses this by using callback-enhanced memory, in which the agent generates a callback query each step to selectively retrieve and revisit information from its previous memories and thereby enabling non-linear reasoning. To train this agent effectively, the authors propose RLMLR and combines sparse outcome-based rewards with dense, step-level rewards based on the "information gain" in memory updates and callback retrievals. Experiments show that the proposed ReMemR1 can outperforms existing memory-based approaches and long-context alternatives.

**Strengths:**

1. The proposed method allows memory agents to selectively retrieve information from its previous memories. This counters the potential information loss in standard memory agent methods and enables non-linear reasoning paths.

2. The propsoed RLMLR introduces a RL-learning framework with multi-level reward design. This provides more dense and step-level rewards for each states in the reasoning trajectory, which offers better guidance than RL methods that solely rely on the final outcome.

3. The authors perform extensive experiments to show that ReMemR1 can outperfor memory agents and lon-context LLMs in open-domain QA benchmarks. They also highlight the performance gains on out-of-distribution datasets.

**Weaknesses:**

1. Limited datasets / baselines. For open-domain QA benchmarks, the authors should also consider one-hop datasets like triviaQA, natural questions and further multi-hop questions like bamboogle, musique. Further long-context baselines like IterDRAG and search agents like Search-R1 should also be considered as baselines.

2. The method requires storing the entire history of all previous memory states over many retrieved document chunks. For each step, the LLM then perofrms a retrieval search across this history and generate new states / memory or final answer. This adds additional compute, memory overhead and latency compared to standard single-pass methods or search agents.

3. The authors adopt a very simple retrieval function based on word overlapping, other alternative methods like embedding-based retrieval is not discussed. Furthermore, the proposed method is designed to return one single past memory with the maximum word overlapping. This could be a limitation preventing the agent from synthesizing information across multiple documents / reasoning steps.

**Questions:**

1. On the RL training level, is each state treated as one training instance in RLMLR?

2. What is the precise format of the memory? Can you provide more qualitative examples of the memories and the generated queries?

---

> ### Author Response · Authors · 2025-11-21
> **Author Response to Reviewer K85P (1/3)**
>
> We thank the reviewer for the time and effort in reviewing this paper.
>
> > **Weakness 1. Limited datasets / baselines.** For open-domain QA benchmarks, the authors should also consider one-hop datasets like triviaQA, natural questions and further multi-hop questions like bamboogle, musique. Further long-context baselines like IterDRAG and search agents like Search-R1 should also be considered as baselines.
>
> Thank you for the reviewer’s suggestion. We agree that adding more baselines and benchmarks can substantially strengthen our paper. Following your advice, we expanded our experiments with **additional baselines and benchmarks**, including:
> - Additional Baselines:
>   - (1) search agent: Search-R1
>   - (2) memory-augmented LLMs: MemOS
> - Additional Benchmarks:
>   - (1) single-hop QA: NQ and TriviaQA
>   - (2) multi-hop QA: Bamboogle and Musique
>
>
> **Table 1.** Performance comparison with **additional baselines**.
>
> ||Number of Docs|50|100|200|400|800|1600|3200|6400|
> |---|---|---|---|---|---|---|---|---|---|
> |HotpotQA|R1-Distill|40.6|25.8|10.2|0.8|1.6|2.3|1.5|3.1|
> ||Qwen2.5-1M|75.8|71.9|68.0|67.2|69.5|54.7|22.7|0.0|
> ||MemAgent|81.8|78.9|78.9|77.0|79.7|72.1|74.0|75.8|
> ||MemOS w/ GPT-4|70.0|62.0|58.8|57.6|68.1|64.8|60.6|63.7|
> ||Search-R1|32.4|38.0|39.3|38.5|41.6|40.3|41.2|40.5|
> ||Ours|82.3|82.8|81.1|78.9|82.0|79.7|80.0|80.8|
> |2WikiMultiHotQA|R1-Distill|36.7|29.7|25.8|0.0|0.8|2.3|2.3|0.8|
> ||Qwen2.5-1M|62.5|59.4|57.8|47.7|46.1|45.3|25.8|0.0|
> ||MemAgent|61.7|57.8|50.8|47.6|50.7|44.5|46.9|44.7|
> ||MemOS|41.5|37.1|33.1|26.8|29.8|29.1|28.6|27.9|
> ||Search-R1|32.6|37.8|39.6|37.0|41.8|40.4|40.7|41.0|
> ||Ours|63.9|63.1|55.6|54.5|54.7|45.4|48.9|50.3|
>
> **Table 2.** Performance comparison on **additional benchmarks**.
>
> | | Number of Docs | 50 | 100 | 200 | 400 | 800 | 1600 | 3200 | 6400 |
> | -- | --- | ---- | ---- | ---- | ---- | ---- | ---- | ---- | ---- |
> | HotpotQA | MemAgent | 81.8 | 78.9 | 78.9 | 77.0 | 79.7 | 72.1 | 74.0 | 75.8 |
> | | ReMemR1 | 82.3 | 82.8 | 81.1 | 78.9 | 82.0 | 79.7 | 80.0 | 80.8 |
> | 2WikiMultiHotQA | MemAgent | 61.7 | 57.8 | 50.8 | 47.6 | 50.7 | 44.5 | 46.9 | 44.7 |
> | | ReMemR1 | 63.9 | 63.1 | 55.6 | 54.5 | 54.7 | 45.4 | 48.9 | 50.3 |
> | BamBoogle | MemAgent | 25.3 | 31.8 | 27.5 | 24.9 | 31.1 | 27.2 | 28.0 | 28.8 |
> | | ReMemR1 | 44.3 | 45.9 | 44.6 | 43.8 | 43.7 | 41.6 | 46.0 | 41.4 |
> | Musique | MemAgent | 9.3 | 9.6 | 12.2 | 16.1 | 10.9 | 12.8 | 14.5 | 13.3 |
> | | ReMemR1 | 20.7 | 18.4 | 15.2 | 25.0 | 15.0 | 16.4 | 16.8 | 18.8 |
> | NQ | MemAgent | 37.8 | 43.1 | 35.6 | 42.7 | 40.5 | 35.9 | 40.1 | 34.8 |
> | | ReMemR1 | 38.4 | 44.2 | 43.9 | 44.2 | 42.4 | 38.7 | 41.9 | 38.2 |
> | TriviaQA | MemAgent | 36.6 | 36.1 | 40.9 | 31.7 | 36.0 | 35.6 | 27.0 | 31.7 |
> | | ReMemR1 | 37.2 | 35.6 | 42.8 | 32.1 | 37.5 | 35.7 | 27.2 | 31.3 |
> | Average | MemAgent | 42.1 | 42.9 | 41.0 | 40.0 | 41.5 | 38.0 | 38.4 | 38.2 |
> | | ReMemR1 | 47.8 | 48.3 | 47.2 | 46.4 | 45.9 | 42.9 | 43.5 | 43.5 |
>
>
> All four newly added benchmarks are padded with random documents, following the same setting as our Table 1 in paper. The results are reported in Table 1 and Table 2. From these results, we observe that:
> 1. ReMemR1 **consistently outperforms** the existing baseline methods on the new benchmarks, and outperforms the newly added baselines on the original benchmarks.
> 2. Table 2 further shows that ReMemR1 exhibits **larger performance gains on multi-hop benchmarks**, which we attribute to its ability to mitigate the *irreversible processing* issue that previous approaches suffer from.
> 3. Search-R1 underperforms memory-based LLMs, while the results align with the ones in their paper. The performance gap are caused by only the retrieved documents are provided to the LLM, but the full context are not exactly visible.
>
> So far, we have not found any open-source implementation of the suggested baseline **IterDRAG**. We are currently working on reproducing this method, and we will include the results once they are ready.

---

> ### Author Response · Authors · 2025-11-21
> **Author Response to Reviewer K85P (2/3)**
>
> > **Weakness 2.** The method requires storing the entire history of all previous memory states over many retrieved document chunks. For each step, the LLM then perofrms a retrieval search across this history and generate new states / memory or final answer. This adds additional compute, memory overhead and latency compared to standard single-pass methods or search agents.
>
> We appreciate the reviewer for highlighting the lack of detailed discussion on time and space overhead in our work—this is indeed an important aspect. To address this, we conducted **additional experiments** on the HotpotQA dataset to compare our method against previous work in terms of inference-time accuracy, **time overhead** (computation + latency), and **space overhead** (memory usage). The results are summarized in Table 3.
>
>
> **Table 3.** Comparison of accuracy, time and space overhead between previous paradigm and our work.
>
> | | | Number of Docs | 50 | 100 | 200 | 400 | 800 | 1600 | 3200 | 6400 |
> | -- | -- | -- | -- | -- | -- | -- | -- | -- | -- | -- |
> | Accuracy | MemAgent | | 81.8 | 78.9 | 78.9 | 77 | 79.7 | 72.1 | 74 | 75.8 |
> | | ReMemR1 | | 82.3 | 82.8 | 81.1 | 78.9 | 82 | 79.7 | 80 | 80.8 |
> |Time|MemAgent|Time / Sample (s)|14.51|22.41|38.16|69.89|152.62|356.82|676.80|1422.17|
> ||ReMemR1|Time / Sample (s)|16.70|26.02|46.90|90.53|211.33|527.85|1004.29|1935.84|
> ||ReMemR1|Callback Time / Sample (s)|0.01|0.03|0.06|0.12|0.27|0.40|0.93|1.73|
> ||ReMemR1|Callback / Total|0.07%|0.10%|0.14%|0.14%|0.13%|0.08%|0.09%|0.09%|
> | Space | MemAgent | Total Memory (MB) | 811.95 | 818.32 | 833.18 | 862.68 | 924.18 | 1050.88 | 1358.79 | 1989.44 |
> | | ReMemR1 | Total Memory (MB) | 808.90 | 824.80 | 868.80 | 893.44 | 964.17 | 1117.41 | 1418.86 | 2005.30 |
> | | ReMemR1 | Callback Memory (MB) | 0.01 | 0.02 | 0.03 | 0.05 | 0.10 | 0.16 | 0.23 | 0.34 |
> | | ReMemR1 | Callback / Total | <0.001% | <0.001% | <0.001% | <0.001% | <0.001% | <0.001% | <0.001% | <0.001% |
>
> To better illustrate our approach, we also provide a theoretical comparison of complexity between our method and prior work, shown in Table 4. Here,
> - $L_{\text{ext}}$ denotes the length of external memory (i.e., context documents, 50, 100, 200, … in our experiments),
> - $L_{\text{int}}$ denotes the length of internal memory (model-generated memory), and
> - $N$ is the number of interaction steps.
>
> **Table 4.** Comparison of between previous paradigm and our work
>
> | | "Memorize while Reading" paradigm | Our Method |
> | --- | --- | --- |
> | Overview |  (1) Process context chunk-by-chunk (2) Maintain only the latest memory state | (1) Process context chunk-by-chunk (2) Maintain *all* memory states (3) Retrieve memories on demand during inference |
> | Benefits | Lower time & memory cost | Addresses irreversible processing and progressive information loss |
> | Drawbacks | Progressive information loss and Irreversible processing | Quadratic computation scaling with respect to $N$ |
> | CPU Computation | $\mathcal{O}\!\left(N \cdot L_{\text{int}} + L_{\text{ext}}\right)$ | $\mathcal{O}\!\left(N^{2} \cdot L_{\text{int}} + L_{\text{ext}}\right)$ |
> | Memory Usage per Sample | $\mathcal{O}(L_{\text{int}} + L_{\text{ext}} / N)$ | $\mathcal{O}(N \cdot L_{\text{int}} + L_{\text{ext}} / N)$ |
> | GPU Computation | $\mathcal{O}\!\left(N \cdot \left(\frac{L_{\text{ext}}}{N} + L_{\text{int}}\right)^2 \, L_{\text{int}}\right)$ | $\mathcal{O}\!\left(N \cdot \left(\frac{L_{\text{ext}}}{N} + L_{\text{int}}\right)^2 \, L_{\text{int}}\right)$ |
>
> In this work, we identify the limitations of prior approaches and propose a lightweight solution to mitigate them. Thanks to the design of our lightweight retriever, the higher-order terms in the complexity expressions are all proportional to $L_{\text{int}}$, which is typically several orders of magnitude smaller than $L_{\text{ext}}$. As shown in Table 3, this tradeoff is worthwhile: our method achieves **up to 5% absolute accuracy improvement**, corresponding to a **20% reduction in error rate**, while incurring only acceptable additional time and memory costs.

---

> ### Author Response · Authors · 2025-11-21
> **Author Response to Reviewer K85P (3/3)**
>
> > **Weakness 3.** The authors adopt a very simple retrieval function based on word overlapping, other alternative methods like embedding-based retrieval is not discussed. Furthermore, the proposed method is designed to return one single past memory with the maximum word overlapping. This could be a limitation preventing the agent from synthesizing information across multiple documents / reasoning steps.
>
>
> We thank the reviewer for the helpful suggestion regarding improving the retrieval module.
>
> Since our model is trained with multi-machine parallelism, we have encountered certain engineering issues when attempting to implement an embedding-based retriever, and these appears to be difficult to resolve within the limited discussion period. Following your advice, we instead replaced our original word-overlapping retriever with a **TF-IDF–based semantic retriever**, and re-trained / re-evaluated our model. The results are shown in Table 5.
>
> **Table 5.** Performance comparison of ReMemR1 with the original word-overlap retriever vs. a TF-IDF semantic retriever.
>
> ||Number of Docs|50|100|200|400|800|1600|3200|6400|
> |---|---|---|---|---|---|---|---|---|---|
> |HotpotQA|ReMemR1|82.3|82.8|81.1|78.9|82.0|79.7|80.0|80.8|
> ||ReMemR1 w/ TF-IDF Retriever|84.2|84.9|84.4|80.5|84.4|79.7|82.8|81.9|
> |2WikiMultiHotQA|ReMemR1|63.9|63.1|55.6|54.5|54.7|45.4|48.9|50.3|
> ||ReMemR1 w/ TF-IDF Retriever|66.4|62.5|57.0|59.4|56.3|49.2|55.8|54.1|
>
> From these results, we can see that ReMemR1 trained and evaluated with the stronger retriever achieves consistently better performance. This confirms that **ReMemR1 indeed benefits from improvements in the retrieval module**.
>
> > **Question 1.** On the RL training level, is each state treated as one training instance in RLMLR?
>
> **Yes.** During training, the model is optimized to generate $m_t$ given the state $s_t$ at each step. Therefore, if there are $N$ rounds of recurrent memory updating (also denoted as $T$ in Figure 3 of the paper) with a batch size of $BSZ$, the total number of training samples used for policy updates becomes $N \times BSZ$.
>
>
> > **Question 2.** What is the precise format of the memory? Can you provide more qualitative examples of the memories and the generated queries.
>
> The memories are free-form text up to 1024 tokens, generated by the LLM. We are **currently preparing an additional case study section** in the appendix to better illustrate both the memory states and the generated callback queries.

---

> ### Author Response · Authors · 2025-11-28
> **New Updates to the paper & Looking forward to discuss with you**
>
> As the discussion period progresses, we want to inform you that we have uploaded a **revised manuscript** containing significant updates and new experimental data. We believe these additions strengthen the paper and may address potential questions regarding efficiency and model behavior.
>
> Summary of Key Updates:
> 1. **Clarified Motivation (Intro & Abstract):** We modified the Introduction and Abstract to better articulate the limitations of forward-only models. We specifically clarify why a callback mechanism is strictly necessary compared to simply improving the memory update policy.
>
> 2. **Computational Overhead Analysis (New RQ3, Section 3.4, Table 5):** We added a detailed efficiency analysis. The results demonstrate that ReMemR1 achieves **significant accuracy gains with negligible overhead**, addressing potential scalability concerns.
>
> 3. **Extended Qualitative Analysis (Appendix B.5 & B.6):** We added **six pages of new case studies and failure analysis**. These visualization examples demonstrate how our method successfully retrieves past evidence where baselines fail, while also transparently discussing remaining limitations.
>
> We are continuing to work on the paper (including new baselines and framework refinements) throughout the rebuttal period. We would value your thoughts on these updates and remain available to answer any questions you may have.
>
> Best regards,
>
> The Authors

---

### Official Review · Reviewer_1HMb · 2025-11-03

**Soundness:** 3
**Presentation:** 3
**Contribution:** 3
**Rating:** 6
**Confidence:** 3

**Summary:**

This paper addresses the challenge of question-answering over long contexts, in which large language models (LLMs) typically utilize a memory corpus to summarize a single pass over the context in order to answer the question. This approach, which uses only forward processing, is storage-efficient but often suboptimal, as the importance of some parts of the context may only be apparent after scanning the initial context. This paper proposes to mitigate this challenge by appending a query and retrieval function to the agent memory for each document chunk. This query can retrieve information from prior chunks, allowing them to be “revisited.” The paper further introduces chunk-level rewards that measure the information gain in memory updates to encourage retention of new information relevant to the ground truth, using reinforcement learning to learn the memories and retrievals for each document chunk. Experiments on multiple short and long context datasets show that the proposed ReMemR1 technique outperforms MemAgent as well as existing standard and long-context LLMs.

**Strengths:**

+ The experiment results show a clear gain in performance relative to the baselines for both in-distribution and out-of-distribution test datasets, for most document context lengths. The accuracy of ReMemR1 is also fairly consistent as the number of context documents increases.

+ The proposed idea of using learned queries to revisit past context information, while simple to conceptualize, is intuitively likely to work. It also appears to use about the same amount of memory as the usual single-pass approaches (though I’d appreciate if the authors could confirm this explicitly).

**Weaknesses:**

--It seems like the presence of learned queries in ReMemR1 would increase the LLM inference time, as the model would need to query past context while making an inference. How much more time would this take in practice (and if I am incorrect and the inference time is comparable to that of single-pass memory agents, can the authors explain why?)

-- The experiments pad the contexts of both HotPotQA and 2WikiMultiHopQA with random documents. This would seem to give an unrealistic picture of true long-context documents, which would very likely have different patterns from random. Indeed, it seems that the random padding would simply lead to the model learning not to query these random documents.

**Questions:**

Please see also Weaknesses above.

1) What is the tradeoff introduced by varying alpha? I understand that a smaller value of alpha corresponds to more emphasis on step-based rewards, but conceptually, since the evaluation target is the final accuracy, the optimal value of alpha should be alpha = 1, which emphasizes only the final reward. Thus, the paper should better justify the need for step-based rewards. Given enough training iterations, will alpha=1 eventually converge to a better (i.e., higher accuracy) solution?

2) Why does ReMemR1 use the relatively simple word similarity as a reward in the step-based rewards? Would a more sophisticated information gain measure (e.g., one that evaluates the semantics) work better?

3) The experiment section also includes an ablation study of sorts that compares ReMemR1 to that of MemAgent with a rule-based callback, showing that ReMemR1’s learned callbacks outperform a rule-based one. It is not clear, however, whether this is a true ablation study on ReMemR1; is MemAgent equivalent to ReMemR1 without revisitable memory? It’s also not clear why MemAgent + rule-based callback often performs worse than MemAgent itself in Table 3.

4) In Figure 5 of Appendix B, it’s not clear why the paper states that performance rapidly improves around epoch 20. There seems instead to be a smoothly decreasing increase in performance across the training duration.

---

> ### Author Response · Authors · 2025-11-21
> **Author Response to Reviewer 1HMB (1/3)**
>
> We thank the reviewer for the time and effort in reviewing this paper.
>
> > **Weakness 1.** It seems like the presence of learned queries in ReMemR1 would increase the LLM inference time, as the model would need to query past context while making an inference. How much more time would this take in practice (and if I am incorrect and the inference time is comparable to that of single-pass memory agents, can the authors explain why?)
>
> We thank the reviewer for highlighting the potential extra latency introduced by ReMemR1 at inference time. To better quantify this overhead, we conduct **additional experiments** measuring the latency of the retrieval process. The results are shown in Table 1.
>
> **Table 1.** Comparison of accuracy and latency between previous paradigm and our work.
>
> | | | Number of Docs | 50 | 100 | 200 | 400 | 800 | 1600 | 3200 | 6400 |
> | -- | -- | -- | -- | -- | -- | -- | -- | -- | -- | -- |
> | Accuracy | MemAgent | | 81.8 | 78.9 | 78.9 | 77 | 79.7 | 72.1 | 74 | 75.8 |
> | | ReMemR1 | | 82.3 | 82.8 | 81.1 | 78.9 | 82 | 79.7 | 80 | 80.8 |
> | Latency | MemAgent | Time / Sample (s)          | 14.51 | 22.41 | 38.16 | 69.89 | 152.62 | 356.82 | 676.80  | 1422.17 |
> |      | ReMemR1  | Time / Sample (s)          | 16.70 | 26.02 | 46.90 | 90.53 | 211.33 | 527.85 | 1004.29 | 1935.84 |
> |      | ReMemR1  | Callback Time / Sample (s) | 0.01  | 0.03  | 0.06  | 0.12  | 0.27   | 0.40   | 0.93    | 1.73    |
> |      | ReMemR1  | Callback / Total | 0.07%|0.10%|0.14%|0.14%|0.13%|0.08%|0.09%|0.09%|
>
> From the empirical results, we observe the following:
> 1. **ReMemR1 does introduce additional time overhead, but the increase is modest**. The callback time consistently stays around 4–6% of the total latency. This indicates that the retrieval over stored memories is quite lightweight in practice.
> 2. Beyond the callback cost, **the majority of ReMemR1’s additional latency comes from the LLM generating retrieval-query tokens at each step**, rather than from the retrieval process itself. This results in the total overhead is slightly higher than the callback component alone.
> 3. In exchange for this small time increase, ReMemR1 delivers up to 5% absolute accuracy improvements, corresponding to a **20% reduction in error rate**. This tradeoff demonstrates that the additional latency is well justified.
>
> > **Weakness 2.** The experiments pad the contexts of both HotPotQA and 2WikiMultiHopQA with random documents. This would seem to give an unrealistic picture of true long-context documents, which would very likely have different patterns from random. Indeed, it seems that the random padding would simply lead to the model learning not to query these random documents.
>
> We understand the reviewer’s concern regarding the potential impact of padding strategy on the reasoning process. To validate this, we conducted **additional experiments** using padding documents that are more relevant to the question, simulating a more realistic long-context setting.
>
> We measure document relevance using the embedding similarity to the golden passages. For example, under the 50-document long-context setting, instead of randomly sampling 50 documents, we first select the top 500 documents most similar (in embedding space) to the golden passages, and then randomly sample 50 documents from this filtered set. We performed this evaluation for both our method and the previous “memorize-while-reading” method, MemAgent. The comparison is shown in Table 2.
>
> **Table 2.** Influence of different padding strategies
>
> |||Number of Docs|50|100|200|400|800|1600|3200|6400|
> |---|---|---|---|---|---|---|---|---|---|---|
> |HotpotQA|Random Padding|MemAgent|81.8|78.9|78.9|77|79.7|72.1|74|75.8|
> |||ReMemR1|82.3|82.8|81.1|78.9|82|79.7|80|80.8|
> ||Padding w/ Relevant Content|MemAgent|85.9|81.6|79.8|77.3|74.2|85.2|81.3|80.5|
> |||ReMemR1|89.8|85.3|83.3|80.9|83.3|88.2|89.6|84.6|
> |2WikiMultiHopQA|Random Padding|MemAgent|61.7|57.8|50.8|47.6|50.7|44.5|46.9|44.7|
> |||ReMemR1|63.9|63.1|55.6|54.5|54.7|45.4|48.9|50.3|
> ||Padding w/ Relevant Content|MemAgent|52.3|60.2|51.6|55.5|52.3|43.0|45.3|48.1|
> |||ReMemR1|57.0|65.6|57.0|60.2|56.3|44.5|46.4|54.0|
>
>
> From the results, we observe:
> 1. For both methods, **padding with more relevant documents generally improves overall performance**, confirming that relevance-based padding provides a more realistic and beneficial evaluation setting.
> 2. Some performance drops do occur, e.g., a 7–8% decrease at 50 documents on 2WikiMultiHopQA. After examining the inference logs, we find that this is caused by the presence of ambiguous documents: *documents that contain content highly similar to the golden passages but do not include the correct final answer. **These documents can mislead the model’s reasoning and memory updates**.

---

> ### Author Response · Authors · 2025-11-21
> **Author Response to Reviewer 1HMB (2/3)**
>
> > **Question 1.** What is the tradeoff introduced by varying alpha? I understand that a smaller value of alpha corresponds to more emphasis on step-based rewards, but conceptually, since the evaluation target is the final accuracy, the optimal value of alpha should be alpha = 1, which emphasizes only the final reward. Thus, the paper should better justify the need for step-based rewards. Given enough training iterations, will alpha=1 eventually converge to a better (i.e., higher accuracy) solution?
>
> We agree that the tradeoff introduced by varying $\alpha$ is an important design aspect. To better illustrate its impact, we included **additional visualizations** comparing training curves under different $\alpha$ values (shown in Figure 6, Appendix B.3 of the revised paper).
>
> From these curves, we observe that using an appropriate $\alpha$ indeed helps the model **converge to a better final outcome reward**. This aligns with the intended-purpose of step-level rewards: by identifying which intermediate steps meaningfully contribute to final success, the step-level signals help alleviate the *sparse and noisy credit assignment* issues present in purely outcome-based rewards. In other words, step-level rewards help the model “memorize better” by reinforcing the useful intermediate reasoning behaviors.
>
> Table 3 in the paper further confirms this: across four fully-converged training runs, **$\alpha = 0.8$ yields stronger evaluation performance** compared to $\alpha = 1$.
>
>
> > **Question 2.** Why does ReMemR1 use the relatively simple word similarity as a reward in the step-based rewards? Would a more sophisticated information gain measure (e.g., one that evaluates the semantics) work better?
>
> We strongly agree with the reviewer’s point in Question 1 that "the evaluation target is the final accuracy, so the training objective should be aligned with it".
>
> Our current design of step-level rewards follows exactly this intuition:
> - The term $r^{(g)}_{memory,t}$ evaluates **whether the words in the answer explicitly appear in the current memory**,
> - The final outcome reward (and the evaluation metric) measures the **explicit occurrence of the correct answer** via exact match accuracy.
> Given this alignment, we believe a word-level indicator is appropriate **as long as the evaluation metric is also word-based**, despite being simpler than semantic-based ones.
>
> > **Question 3.** The experiment section also includes an ablation study of sorts that compares ReMemR1 to that of MemAgent with a rule-based callback, showing that ReMemR1’s learned callbacks outperform a rule-based one. It is not clear, however, whether this is a true ablation study on ReMemR1; is MemAgent equivalent to ReMemR1 without revisitable memory? It’s also not clear why MemAgent + rule-based callback often performs worse than MemAgent itself in Table 3.
>
>
> The reviewer raises a very fair concern regarding our ablation design. We agree that the previous Table 3 was not a proper ablation study, but more of an analytical comparison.
>
> To address this, we conducted an **additional ablation study**, evaluating:
> 1. vanilla ReMemR1,
> 2. ReMemR1 **without callback**, and
> 3. ReMemR1 with the callback queries replaced by the **original question** (instead of LLM-written queries).
>
> The results are shown in Table 3.
>
> **Table 3.** Ablation study over the callback mechanism on HotpotQA
>
> |Number of Docs|50|100|200|400|800|1600|3200|6400|
> |---|---|---|---|---|---|---|---|---|
> |ReMemR1|82.3|82.8|81.1|78.9|82|79.7|80|80.8|
> |- w/o LLM-written Callback Query|81.9|82.5|80.3|77.7|75.5|79.8|77.8|78.6|
> |- w/o Memory Callback|82.3|79.7|80.5|78.1|80.6|78.6|77.8|78.1|
>
> We see that:
> 1. Both new settings cause performance degradation, especially at larger context lengths (≥ 800 docs)
> 2. These gaps widen as the number of documents goes up.
> 3. "w/o LLM-written Callback Query" results in larger drop than "w/o Memory Callback" variant on 200~800 docs, which can be attributed to the fact that poorly targeted retrieval signals are more harmful than having no retrieval at all.
>
> >  It’s also not clear why MemAgent + rule-based callback often performs worse than MemAgent itself in Table 3.
>
> Regarding “why MemAgent + rule-based callback sometimes performs worse than vanilla MemAgent,” our interpretation is as follows:
> - MemAgent was never trained to use recalled memory;
> - When given rule-based callbacks, it occasionally over-relies on the recalled memory, neglecting the current document context;
> - As the context length grows, the usefulness of rule-based callbacks becomes more significant, resulting in its performance gradually surpasses the vanilla MemAgent when context length grows beyond 1.6k documents.

---

> > ### Author Response · Authors · 2025-11-21
> > **Author Response to Reviewer 1HMB (3/3)**
> >
> > > **Question 4.** In Figure 5 of Appendix B, it’s not clear why the paper states that performance rapidly improves around epoch 20. There seems instead to be a smoothly decreasing increase in performance across the training duration.
> >
> > We thank the reviewer for pointing this out. The original statement was indeed unclear. The correct statement should be:
> >
> > *“performance rapidly improves **during the first 20 steps**”*
> >
> > This reflects that the **format reward sharply improves the model’s formatting ability during the early training phase**, after which the performance changes more gradually. We have corrected this phrasing in the revised version.

---

> ### Author Response · Authors · 2025-11-28
> **New Updates to the paper & Looking forward to discuss with you**
>
> As the discussion period progresses, we want to inform you that we have uploaded a **revised manuscript** containing significant updates and new experimental data. We believe these additions strengthen the paper and may address potential questions regarding efficiency and model behavior.
>
> Summary of Key Updates:
> 1. **Clarified Motivation (Intro & Abstract):** We modified the Introduction and Abstract to better articulate the limitations of forward-only models. We specifically clarify why a callback mechanism is strictly necessary compared to simply improving the memory update policy.
>
> 2. **Computational Overhead Analysis (New RQ3, Section 3.4, Table 5):** We added a detailed efficiency analysis. The results demonstrate that ReMemR1 achieves **significant accuracy gains with negligible overhead**, addressing potential scalability concerns.
>
> 3. **Extended Qualitative Analysis (Appendix B.5 & B.6):** We added **six pages of new case studies and failure analysis**. These visualization examples demonstrate how our method successfully retrieves past evidence where baselines fail, while also transparently discussing remaining limitations.
>
> We are continuing to work on the paper (including new baselines and framework refinements) throughout the rebuttal period. We would value your thoughts on these updates and remain available to answer any questions you may have.
>
> Best regards,
>
> The Authors

---

### Author Response · Authors · 2025-12-02
**Summary of rebuttal period, reviewers' opinions, and authors' responses to address the concerns**

Dear Area Chair,

Our paper received an original score of **6644**.

During the rebuttal period, we have fully addressed the concerns of Reviewer $\textcolor{red}{\textbf{1HMb}}$ (6) and Reviewer $\textcolor{green}{\textbf{EGXY}}$ (6), both of whom **kept their scores** without further replies.
Reviewer $\textcolor{orange}{\textbf{VkNG}}$ (4) actively engaged in the rebuttal and expressed an intention to **raise the score**.

Reviewer $\textcolor{blue}{\textbf{K85P}}$ (4) has not replied yet, but acknowledged our novelty and strong performance. The reviewer's concerns are about additional empirical results:
1. **Additional baseline method.** Following the reviewer’s suggestion, we added Search-R1 and MemOS as baseline methods.
2. **Additional benchmarks.** We additionally compared performance on the four suggested datasets: NQ, TriviaQA, Bamboogle, and Musique.
3. **Computational overhead.** We conducted theoretical analyses and additional experiments evaluating the time and memory complexity of our method.
4. **Alternative retriever designs.** We conducted new experiments replacing the word-overlap retriever with a TF-IDF semantic retriever.

Below, we would like to summarize the **strengths** acknowledged by the reviewers, the **major concerns**, and **our responses**.

## Strengths Acknowledged by Reviewers

1. **Strong Performance:** Reviewers highlighted our “extensive experiments,” showing “consistent improvements” and “clear gains” relative to baselines (Reviewers $\textcolor{red}{\textbf{1HMb}}$, $\textcolor{green}{\textbf{EGXY}}$, $\textcolor{blue}{\textbf{K85P}}$).
2. **Methodological Novelty:** Reviewers found the callback mechanism “conceptually interesting” and “intuitively likely to work,” and the "better guidance" or our RL algorithm (Reviewers $\textcolor{red}{\textbf{1HMb}}$, $\textcolor{orange}{\textbf{VkNG}}$, $\textcolor{blue}{\textbf{K85P}}$).
3. **Clear Motivation & Writing:** The paper is described as “clearly written, well-organized, and easy to follow,”. (Reviewer $\textcolor{orange}{\textbf{VkNG}}$).

## Major Concerns

1. **Computational Overhead:** Questions regarding latency, memory usage, and the scalability of storing history (Reviewers $\textcolor{red}{\textbf{1HMb}}$, $\textcolor{blue}{\textbf{K85P}}$, $\textcolor{green}{\textbf{EGXY}}$, $\textcolor{orange}{\textbf{VkNG}}$).
2. **Baselines & Benchmarks:** Requests for additional baselines (e.g., Search-R1, MemOS, by Reviewers $\textcolor{blue}{\textbf{K85P}}$, $\textcolor{orange}{\textbf{VkNG}}$), and more benchmarks (e.g., NQ, TriviaQA, Bamboogle, and Musique, by Reviewer $\textcolor{blue}{\textbf{K85P}}$).
3. **Robustness under Alternative Designs:** Concerns about random padding strategies (Reviewer $\textcolor{red}{\textbf{1HMb}}$) and the robustness of the retrieval mechanism (Reviewer $\textcolor{blue}{\textbf{K85P}}$).
4. **Theoretical Grounding:** Requests for complexity analysis and justification of the reward design (Reviewers $\textcolor{red}{\textbf{1HMb}}$, $\textcolor{orange}{\textbf{VkNG}}$).

## Our Responses

Our responses to address these concerns are summarized below:

1. **Efficiency Verification:** In response to Reviewer $\textcolor{blue}{\textbf{K85P}}$’s concern regarding **computational overhead**, we conducted **latency and memory tests** and provided a Big-O complexity comparison. Results show that ReMemR1 achieves a 20% reduction in error rate with negligible overhead.
2. **Expanded Benchmarking:** In response to Reviewer $\textcolor{blue}{\textbf{K85P}}$’s concern regarding **baselines and benchmarks**, we added the **additional baselines** (Search-R1, MemOS) and expanded evaluation to the **four suggested benchmarks** (NQ, TriviaQA, Bamboogle, Musique). Our paper now includes experiments on 2 single-hop QA benchmarks (NQ, TriviaQA) and 4 multi-hop QA benchmarks (HotpotQA, 2WikiMultihopQA, Bamboogle, Musique), comparing against 9 long-context LLMs (including Qwen and DeepSeek models) and 3 memory-augmented LLMs (MemOS, Search-R1, MemAgent).
3. **Qualitative Analysis of Robustness:** In response to Reviewer $\textcolor{blue}{\textbf{K85P}}$’s concern regarding **robustness under alternative designs**, we replaced the word-overlap retriever with a **TF-IDF semantic retriever**, which further improved performance.
4. **Theoretical Justification:** In response to concerns regarding **theoretical grounding**, we provided a Big-O complexity comparison and additional ablations (e.g., removing the LLM-written callback) to justify our architectural choices. We also clarified that “RLMLR” is a specialized reward design tailored for recurrent-state structures.

All major concerns have been fully addressed during the rebuttal period, and we have refined our manuscript with the additional results (highlighted in blue in the PDF). We hope that these improvements will be taken into consideration.

We sincerely appreciate your valuable time and patience.

Thanks and regards,

The Authors

---

### Meta-Review · Area_Chair_kRNo · 2026-01-07

**Summary:**

This paper proposes ReMemR1, a memory-augmented LLM agent with callback-based revisitable memory and a multi-level reinforcement learning reward design (RLMLR) for long-context question answering. Reviewers generally found the motivation clear and the method conceptually interesting, with strong and consistent empirical gains over existing memory agents and long-context baselines. Positive feedback emphasized the novelty of non-linear memory revisiting, extensive experiments, and improved robustness on long and multi-hop QA tasks. Main concerns focused on computational and memory overhead, sufficiency of baselines, clarity of theoretical justification, and presentation issues. The authors provided thorough rebuttals, adding new baselines, benchmarks, efficiency analyses, ablations, and clarifications, which addressed most concerns. Overall recommendation: Accept.

**Reviewer Concerns:**

Reviewer Concerns Addressed by the Rebuttal

1. Computational and memory overhead: Reviewers’ concerns about scalability, latency, and memory usage were largely addressed through new empirical measurements and theoretical complexity analyses, showing modest overhead relative to performance gains.
2. Insufficient baselines and benchmarks: The authors added multiple requested baselines (e.g., Search-R1, MemOS) and new datasets (NQ, TriviaQA, Bamboogle, Musique), substantially strengthening the empirical evaluation.
3. Retrieval and reward design clarity: Additional ablations, alternative retrievers (TF-IDF), and clearer explanations of the multi-level reward design helped justify key architectural choices.

Outstanding or Partially Addressed Concerns

1. Long-term memory growth management: While overhead was quantified, the method still lacks explicit pruning or compression strategies for unbounded memory growth.
2. Theoretical depth: Despite added analyses, some reviewers still view the theoretical understanding of why callbacks outperform improved memory updates as limited.
3. Presentation quality: Figure clarity and organization were improved, but presentation issues were initially significant and may still require careful final polishing.

**Reviewer Scores:**

Reviewer 1HMb: Likely to increase slightly. Most technical questions on latency, padding strategy, rewards, and ablations were carefully addressed with new experiments and clarifications.

Reviewer K85P: Likely to increase to borderline accept or weak accept. Added baselines, datasets, efficiency analysis, and alternative retrievers directly addressed core concerns.

Reviewer EGXY: Likely to maintain or slightly increase. Memory efficiency, reward clarification, and failure analysis were substantially improved, though presentation concerns were secondary.

Reviewer VkNG: Explicitly indicated a score increase after rebuttal, as major concerns about novelty, scalability, and overhead were resolved to their satisfaction.

---

### Decision · Program_Chairs · 2026-01-26

Accept (Poster)